# CrocO_v1.0: a Particle Filter to assimilate snowpack observations in a spatialised framework

Bertrand Cluzet[1], Matthieu Lafaysse[1], Emmanuel Cosme[2], Clément Albergel[3], Louis-François Meunier[3], and Marie Dumont[1]

[1]Univ. Grenoble Alpes, Université de Toulouse, Météo-France, CNRS, CNRM, Centre d'Études de la Neige, Grenoble, France
[2]Institut des Géosciences de l'Environnement, IGE, UGA-CNRS, Grenoble, France
[3]CNRM, Université de Toulouse, Météo-France, CNRS, 31057 Toulouse, France.

**Correspondence:** Bertrand Cluzet (bertrand.cluzet@meteo.fr)

**Abstract.** Monitoring the evolution of snowpack properties in mountainous areas is crucial for avalanche hazard forecasting and water resources management. In-situ and remotely sensed observations provide precious information on the state of the snowpack but usually offer a limited spatio-temporal coverage of bulk or surface variables only. In particular, visible-near infrared (VIS-NIR) reflectance observations can provide information about the snowpack surface properties but are limited by terrain shading and clouds. Snowpack modelling enables the estimation of any physical variable, virtually anywhere, but is affected by large errors and uncertainties. Data assimilation offers a way to combine both sources of information, and to propagate information from observed areas to non observed areas. Here, we present CrocO, (Crocus-Observations) an ensemble data assimilation system able to ingest any snowpack observation (applied as a first step to the height of snow (HS) and VIS-NIR reflectances) in a spatialised geometry. CrocO uses an ensemble of snowpack simulations to represent modelling uncertainties, and a Particle Filter (PF) to reduce them. The PF is prone to collapse when assimilating too many observations. Two variants of the PF were specifically implemented to ensure that observations information is propagated in space while tackling this issue. The *global* algorithm ingests all available observations with an iterative inflation of observation errors, while the *klocal* algorithm is a localised approach performing a selection of the observations to assimilate based on background correlation patterns. Feasibility-testing experiments are carried out in an identical twin experiment setup, with synthetic observations of HS and VIS-NIR reflectances available in only a 1/6[th] of the simulation domain. Results show that compared against runs without assimilation, analyses exhibit an average improvement of snow water equivalent Continuous Rank Probability Score (CRPS) of 60% when assimilating HS with a 40-member ensemble, and an average 20% CRPS improvement when assimilating reflectance with a 160-member ensemble. Significant improvements are also obtained outside the observation domain. These promising results open a way for the assimilation of real observations of reflectance, or of any snowpack observations in a spatialised context.

*Copyright statement.* TEXT

# 1 Introduction

Seasonal snowpack is an essential element of mountainous areas. Monitoring the evolution of its physical properties is essential to forecast avalanche hazard (Morin et al., 2020), rain-on-snow related floods (Pomeroy et al., 2016; Würzer et al., 2016) and to monitor water resources (Mankin et al., 2015). Observations alone are too scarce to monitor snowpack conditions. In-situ observations provide precise observations of several key variables, but they lack spatial representativeness and have a poor spatial coverage. Remote sensing of snowpack variables such as the height of snow (HS, (m)), snow water equivalent (SWE, $(\mathrm{kg\,m^{-2}})$), visible-near infrared (VIS-NIR) reflectance, or surface temperature, provide comprehensive information over large areas but usually have a limited temporal resolution on a small set of variables. Furthermore, these observations are usually available in fractions of simulation domains only, even for space-borne data (Davaze et al., 2018; Veyssière et al., 2019; Shaw et al., 2019). For instance, snowpack VIS-NIR reflectances from moderate resolution (250-500 m) satellites such as MODIS or Sentinel-3 can help constraining the snowpack surface properties such as microphysical properties (characterized by the specific surface area, SSA $(\mathrm{m^2kg^{-1}})$ and light absorbing particles content (LAP, $(\mathrm{gg_{snow}^{-1}}))$ (Durand and Margulis, 2006; Dozier et al., 2009). However, in the areas covered by clouds, forests, or concerned by high sub-pixel variability (ridges, roughness, fractional snow cover) and shadows, satellite retrievals are less accurate (Masson et al., 2018; Lamare et al., 2020), and data should be filtered out (Cluzet et al., 2020). The higher resolution offered by products from Landsat or Sentinel-2 might be an avenue to this issue (e.g. Masson et al., 2018; Aalstad et al., 2020) but at these resolution, reflectance retrievals are quite noisy due to e.g. digital elevation model errors (Cluzet et al., 2020). Finally, note that pixel fractional snow cover (snow cover fraction, SCF) can be accurately retrieved even from noisy reflectances (Sirguey et al., 2009; Aalstad et al., 2020), but it inherits its spatio-temporal limitations. SCF informativeness is also limited in deep snowpack conditions (De Lannoy et al., 2012).

Snowpack models of different complexity offer an exhaustive spatial and temporal coverage (Krinner et al., 2018). They are applied within several spatial configurations, including collection of points, regular or irregular grids (Morin et al., 2020). In this paper, "spatialised" refers indistinctly to any of these configurations. Detailed snowpack models are the only ones able to assess avalanche hazard and monitor water resources alike (Morin et al., 2020), but these applications are limited by their considerable errors and uncertainties (Essery et al., 2013; Lafaysse et al., 2017). In that context, combining remote sensing observations with models through data assimilation is an appealing solution (Largeron et al., 2020). Indeed, data assimilation combines the spatial and temporal coverage of snowpack models with the available information from observations in an optimal way. Assimilation of optical reflectance could reduce modelled SWE errors by up to a factor of two (Durand and Margulis, 2007; Charrois et al., 2016), and preliminary studies showed its potential for spatialised assimilation (Cluzet et al., 2020). Assimilation of HS is very efficient in reducing modelled SWE errors (Margulis et al., 2019). However, the limited spatial coverage of observations is stressing the need for data assimilation algorithms able to propagate the snowpack observations information into the unobserved areas (Winstral et al., 2019; Cantet et al., 2019; Largeron et al., 2020).

The Particle Filter with sequential importance resampling (PF-SIR, Gordon et al., 1993; Van Leeuwen, 2009) is a Bayesian ensemble data assimilation technique well suited to snowpack modelling (Dechant and Moradkhani, 2011; Charrois et al., 2016; Magnusson et al., 2017; Piazzi et al., 2018; Larue et al., 2018). The PF-SIR is a sequential algorithm relying on an ensemble of model runs (particles) which represents the forecast uncertainty. At each observation date, the prior (or background) composed of the particles is evaluated against the observations. The analysis of the PF-SIR (later on "PF") works in two steps.

In a first step, so-called "importance sampling", the particles are weighted according to their distance to the observations (relative to the observation errors). Then, a resampling of the particles is performed in order to reduce the variance in the weights. The Ensemble Kalman Filter (EnKF Evensen, 2003), has also been widely used for snow cover data assimilation (e.g. Slater and Clark, 2006; De Lannoy et al., 2012; Magnusson et al., 2014). However, the PF is more adapted to models with a variable number of numerical layers such as detailed snowpack models (Charrois et al., 2016).

The PF could be used in a spatialised context to propagate the information from observation as suggested by Largeron et al. (2020) and Winstral et al. (2019). Contrary to the EnKF, such applications are rare to date (e.g. Thirel et al., 2013; Baba et al., 2018; Cantet et al., 2019). Indeed, spatialised data assimilation with the PF is not straightforward because of the degeneracy issue, i.e. only a few particles are replicated in the analysis, often resulting in a poor representation of the forecast uncertainties. Degeneracy can be mitigated by increasing the number of particles, but the required population scales exponentially with the

number of observations simultaneously assimilated (Snyder et al., 2008). Furthermore, an accurate representation of spatial error statistics by the ensemble is essential for the success of the assimilation system. To achieve that, the required ensemble size also scales exponentially with the system dimension, an issue known as the curse of dimensionality (Bengtsson et al., 2008). These issues are severe drawbacks when considering applications of the PF on large domains (i.e. implying a large number of observations and/or simulation points) with a reasonable number of particles (Stigter et al., 2017).

Several solutions exist to tackle the PF degeneracy. A first approach is to inflate the observation errors in the PF. The tolerance of the PF is increased, leading to more particles being replicated. This approach is based on the fact that observation error statistics (including sensor, retrieval and representativeness errors) are usually poorly known and underestimated. It can also be used as a safeguard to prevent the PF to degenerate on specific dates, when observations are not compatible with the ensemble. PF inflation was successfully implemented in point scale simulations of the snowpack (Larue et al., 2018). When dealing with

a large number of observations, inflation might lead to degeneracy or null analysis (posterior equal to the prior). In this work, we generalize over space the inflation of Larue et al. (2018), trying to ingest all the observations into a single analysis over the domain, in a so-called *global* approach.

PF localisation is a more widespread alternative, tackling degeneracy by reducing the number of observations that are simultaneously assimilated by the PF (Poterjoy, 2016; Poterjoy and Anderson, 2016; Penny and Miyoshi, 2016; Poterjoy et al.,

2019, *italic* notations are taken from the review of Farchi and Bocquet, 2018). In this method, the simulation domain is divided into *blocks* where different PF analyses are performed considering a local subset of observations (*domain*) based on a localisation radius. This makes it possible to constrain the model in locations that are not directly observed, but with nearby observations. Contrary to *global* approaches, localisation has the disadvantage of producing spatially discontinuous analyses (each point receives a different analysis). This issue can be mitigated in various ways (Poterjoy, 2016; Farchi and Bocquet,

2018; Van Leeuwen et al., 2019).

The underlying hypothesis of localisation is that model points are independent beyond a certain distance, i.e. constraining one point with the observation from a too distant point would be meaningless, and likely degrade the analysis performance (Houtekamer and Mitchell, 1998). However, in the case of small simulation domains or modelled systems driven by large-scale coherent causalities, large scale correlations (relative to the domain size) may be physically sound, and defining a localisation

radius may be a difficult task. In order to face this issue, we developed a new localisation approach called the k-localisation, where localisation *domains* are based on background correlation patterns.

These developments were implemented into CrocO (Crocus-Observations) an ensemble data assimilation system able to sequentially assimilate snowpack observations with a PF in a spatialised context. CrocO can be implemented in any geometry, (e.g. within a distributed (gridded) framework or any irregular spatial discretisation). Here, we apply CrocO in a semi-

distributed framework, which is a conceptual spatialised geometry, used operationally by Météo-France for avalanche hazard forecasting (Lafaysse et al., 2013; Morin et al., 2020). This framework is similar to many topographic-based discretisation in hydrological models (e.g. Clark et al., 2015). This setup enables us to account for the snowpack variability induced by the topography at the scale of a mountain range, through meteorological conditions (elevation controls the air temperature and precipitation phase) and the snowpack radiative budget (also dependent on the aspect and slope angle) (Durand et al., 1993).

CrocO uses an ensemble of stochastic perturbations of SAFRAN meteorological analysis (Durand et al., 1993; Charrois et al., 2016) to force ESCROC (Ensemble System CROCus, Lafaysse et al., 2017), the multi-physical version of Crocus snowpack model (Vionnet et al., 2012). The ensemble setup accounts for the major sources of uncertainties in snowpack modelling (Raleigh et al., 2015) and was formerly described and evaluated in the semi-distributed geometry by Cluzet et al. (2020).

Inflation and k-localisation were implemented into CrocO. Here, we present CrocO and evaluate how it addresses the issues

of reflectance observation sparseness and PF degeneracy in the context of snowpack modelling. This problem is divided into two scientific questions: (1) Is CrocO PF able to efficiently spread the information from sparse observations in space without degenerating? (2) Is the spatial information content of reflectance observations valuable for snowpack models? We assess these questions by evaluating the performance of CrocO to model the SWE when assimilating synthetic observations of HS and reflectance covering only a portion of the domain.

Section 2 presents the CrocO system, i.e. the ensemble modelling system and the PF algorithms. Section 3 introduces the evaluation methodology. Subsequently, Sec. 4 assesses the performance of CrocO and Sec. 5 discusses the results. Finally, Sec. 6 provides perspectives and research directions.

## 2   Material and methods

### 2.1   Modelling geometry

Simulations are performed in the semi-distributed geometry. Mountain ranges such as the Alps are discretized into so-called *massifs* of about 1000 km$^2$ to account for regional variability of meteorological conditions. Within each massif, topographic-induced variability is taken into account by running the model for a fixed set of topographic classes, e.g. by 300 m elevation

bands, for 0º, 20º and 40º slopes and 8 aspects (see Fig. 1). This set enables us to reproduce the main features of snowpack variability (e.g. Mary et al., 2013).

In this study, we focus on the Grandes Rousses, a single massif in the central French Alps. This area of about 500 $km^2$ is represented by $N_{pts} = 187$ independent topographic classes (see Fig. 1). In the following, specific topographic classes are denoted as follows: *elevation_aspect_slope*, e.g. 1800_N_40 stands for a 40º slope, with a northern aspect at 1800 m.a.s.l..

## 2.2 CrocO Ensemble data assimilation setup

The ensemble data assimilation workflow of CrocO is represented in Fig. 2. In the following, only a short description of the

system and its elements is provided. More details on the ensemble modelling setup are available in Cluzet et al. (2020). Information about its implementation into Météo-France HPC system can be found in Appendix B1.

### 2.2.1 Ensemble of snowpack models

Crocus is a detailed snowpack model, coupled with the ground and atmosphere in the ISBA land surface model (Interaction

Soil-Biosphere-Atmosphere). It is embedded within the SURFEX_v8.1 modelling platform (SURFace EXternalisée, Masson et al., 2013). The TARTES optical scheme (Libois et al., 2013, 2015) represents VIS-NIR spectral radiative transfer within the snowpack, driven by snow metamorphism (Carmagnola et al., 2014) and Light Absorbing Particles (LAP ($gg_{snow}^{-1}$)) deposition fluxes (Tuzet et al., 2017). Moreover, TARTES computes the snowpack reflectance with a high spectral resolution, making the model directly comparable to the observations. As such, TARTES is both a physical component of Crocus and an observation

operator.

ESCROC (Ensemble System CROCus, Lafaysse et al., 2017) multi-physical ensemble version of Crocus is used to account for snowpack modelling uncertainties. A random draw among 1944 ESCROC multi-physics configurations was performed and used in all the simulations and denoted $(M_i)_{0<i\leq N_e}$, $N_e$ being the ensemble size (e.g. 40 or 160 members, see Fig. 2). These configurations are considered equiprobable before any data assimilation.


### 2.2.2 Ensemble of meteorological forcings

Meteorological forcings are taken from SAFRAN (Durand et al., 1993) reanalysis, where forecasts from the ARPEGE Numerical Weather Prediction (NWP) model are downscaled and adjusted with surface observations within the massif area. They are combined with MOCAGE LAP fluxes (Josse et al., 2004) interpolated at Col du Lautaret (2058 m.a.s.l, inside the Grandes-

Rousses) to constitute the reference forcing dataset. Before the beginning of the simulation, spatially homogeneous stochastic perturbations (e.g. at a given date, the same perturbation parameter is applied across the whole domain) with temporal auto-correlations are applied to this forcing to generate an ensemble of forcings $(F_i)_{0<i\leq N_e}$ with the same procedure as described in Cluzet et al. (2020). More details on the perturbations procedure can be found in Appendix A. At the beginning of the

simulation, each forcing $F_i$ is associated with a random $M_i$ ESCROC configuration and this relation is fixed during the whole

simulation.

### 2.2.3 The Particle Filter in CrocO

The PF is applied sequentially at each observation date on the background state vectors (soil and snowpack state variables, denoted *BG* on Fig. 2). Its analysis is an ensemble of initial conditions used to propagate the model forward. The algorithm

is implemented into SODA ((SURFEX Offline Data Assimilation, Albergel et al., 2017), the data assimilation module of SURFEX_v8.1, enabling a continuous execution sequence between ensemble propagation and analysis, as depicted in Fig. 2.

### 2.3 The Particle Filter equations

At a given observation date, we consider a set of observed variables available at several locations, totalling $N_y$ different observations.

– Each member $\boldsymbol{x}_b^i$ of the background state $\mathbf{X}_b$ is projected into the observation space using the observation operator $h$. In our case, $h$ is just an orthogonal projection on the $N_y$ observations since HS and reflectance are diagnosed within Crocus (see Sec. 2.2.1). The projection $\widehat{\boldsymbol{x}}_b^i = h\boldsymbol{x}_b^i = (\widehat{\boldsymbol{x}}_k^i)_{0<k\leq N_y}$, corresponds to the modelled values at each observed variable/point.

– these $N_y$ observations are collected in the vector $\boldsymbol{y} = (y_k)_{0<k\leq N_y}$. The associated observation error covariance matrix

$\mathbf{R}$ (Eq. 1) is diagonal (e.g. observation errors are assumed independent):

$$\mathbf{R} = \mathbf{diag}(\sigma_k^2, 0 < k \leq N_y) \tag{1}$$

Where $\sigma_k^2$ stands for the observation error variance of observation $k$ and depends only on the type of observation $y_k$ (e.g. HS or reflectance).

The PF analysis usually works in two steps.

– (1) computing the particle weights $w^i$ as the normalised observation likelihood for each particle (Eq. 2):

$$w^i = \frac{e^{-\frac{1}{2}(\boldsymbol{y}-\widehat{\boldsymbol{x}}_b^i)^T \mathbf{R}^{-1}(\boldsymbol{y}-\widehat{\boldsymbol{x}}_b^i)}}{\sum_{k=1}^{N_e} e^{-\frac{1}{2}(\boldsymbol{y}-\widehat{\boldsymbol{x}}_b^k)^T \mathbf{R}^{-1}(\boldsymbol{y}-\widehat{\boldsymbol{x}}_b^k)}} \tag{2}$$

– (2) resampling the particles based on their weights to build the analysis vector $\mathbf{X}_a$. Here, we apply the PF resampling from Kitagawa (1996) which returns $\boldsymbol{s} = (s_i)_{0<i\leq N_e}$, ($s_i \in [1..N_e]$) a sorted vector with duplications, representing the particles to replicate.

A sample reordering step was added for numerical optimisation with no expected incidence on the PF behaviour (see in Appendix B2 for more details).

Two simple variants of this algorithm can be identified in a spatialised context:

– *global* approach: perform one analysis over the domain, putting all the available observations in $\boldsymbol{y}$.

– *rlocal* approach: perform one analysis per model point, assimilating only local observations, if any. This corresponds to a localised PF with *block* and *domain* size of 1.

### 2.3.1 Particle Filter degeneracy

Degeneracy occurs when only a small fraction of the particles have non-negligible weights, resulting in a sample $\boldsymbol{s}$ where only a few different indices are present. It can be diagnosed from the weights using the effective sample size $N_{\mathrm{eff}}$ (Liu and Chen, 1995):

$$N_{\mathrm{eff}} = \frac{1}{\sum_{i=1}^{N_{\mathrm{e}}} (w^i)^2} \tag{3}$$

With a degenerate sample, $N_{\mathrm{eff}} \gtrsim 1$, and with innocuous analysis (all particles are replicated) $N_{\mathrm{eff}} = N_{\mathrm{e}}$.

A first approach to mitigate degeneracy is to use inflation. This heuristic method iteratively inflates $\mathbf{R}$ values until the effective sample size is large enough. Here, we develop a variant from Larue et al. (2018) method, which was not explicitly relying on $N_{\mathrm{eff}}$ (Eq. 3). Consider applying an inflation factor $\frac{1}{\alpha}$ to $\mathbf{R}$, ($0 < \alpha \leq 1$, $\alpha = 1$ being the value for no inflation), and update $N_{\mathrm{eff}}$ (Eqs. 2 and 3): $N_{\mathrm{eff}}$ is naturally a decreasing function of $\alpha$ (the more we inflate $\mathbf{R}$ the more different particles will be replicated). The idea of our method is to ensure that $N_{\mathrm{eff}}$ exceeds a target value, $N_{\mathrm{eff}}^*$. If $N_{\mathrm{eff}} < N_{\mathrm{eff}}^*$ (degenerate case), we reduce $\alpha$ (inflate) until $N_{\mathrm{eff}} = N_{\mathrm{eff}}^*$ using Alg. 1. In the following, inflation is used in the *global* and *rlocal* PF (see Sec. 2.2.3).

The core of Alg. 1 is an hybrid bisection-secant method to find the zero of $f : \alpha \mapsto N_{\mathrm{eff}}(\alpha) - N_{\mathrm{eff}}^*$ in $[0, 1]$. It is inspired by **rtsafe** algorithm (Vetterling et al., 1992). The **guess** function computes a new guess $\alpha_2$ to minimize f. Note that in the unlikely case where Alg. 1 does not converge, all the particles are replicated.

### 2.3.2 k-localisation

In the k-localisation algorithm, degeneracy is mitigated by reducing the number of observations that are simultaneously assimilated. The PF analysis is applied to each simulation point sequentially. In order to build the analysis at point $n$, background correlations $\mathbf{B}_{\boldsymbol{v}}$ are computed for each variable $v$ (e.g HS or reflectance) between $n$ and all the observed points. In a first step, all observations from points exhibiting substantial background correlations (see below **select_k_biggest** function) are used. If the PF degenerates, the number of observations is progressively decreased until degeneracy is mitigated. As earlier, degeneracy is considered mitigated when $N_{\mathrm{eff}} \geq N_{\mathrm{eff}}^*$. This way, we ensure that a maximal number of observations has been ingested by the PF without degenerating.

In case of degeneracy, the observation point displaying the lowest correlation is ruled out. The PF weights are computed (Eq. 2), and a new effective sample size is derived (Eq. 3). While the target sample size is not exceeded, this selection proceeds iteratively. The notation $k$ in "$k$-localisation" refers to the number $k$ of retained observations of each variable. This approach

---

**Algorithm 1** Weighting algorithm with inflation.

---

**Input:** $\widehat{\boldsymbol{x}}^i, \boldsymbol{y}, \mathbf{R}, N_{\text{eff}}^*$

**Output:** $w^i$

1: $\alpha \leftarrow 1$

2: $\mathbf{R} \leftarrow \frac{1}{\alpha}\mathbf{R}$

3: $w^i \leftarrow$ **weights**$(\widehat{\boldsymbol{x}}^i, \boldsymbol{y}, \mathbf{R})$ (Eq. 2)

4: $N_{\text{eff}} \leftarrow$ **eff_weights**$(w^i)$ (Eq. 3)

5: **if** $N_{\text{eff}} < N_{\text{eff}}^*$ **then**

6:     $\alpha_1 \leftarrow 0$

7:     $N_{\text{eff}_1} \leftarrow N_{\text{e}}$

8:     $cond \leftarrow$ **True**

9:     $i \leftarrow 0$

10:    **while** $cond$ **do**

11:        $\alpha_2 \leftarrow$ **guess**$(\alpha_1, \alpha, N_{\text{eff}_1}, N_{\text{eff}}, N_{\text{eff}}^*)$

12:        $\mathbf{R} \leftarrow \frac{1}{\alpha_2}\mathbf{R}$

13:        $w_2^i \leftarrow$ **weights**$(\widehat{\boldsymbol{x}}^i, \boldsymbol{y}, \mathbf{R})$ (Eq. 2)

14:        $N_{\text{eff}_2} \leftarrow$ **eff_weights**$(w_2^i)$ (Eq. 3)

15:        **if** $|N_{\text{eff}_2} - N_{\text{eff}}^*| < \epsilon$ **then**

16:            $cond \leftarrow$ **False**

17:            $\alpha \leftarrow \alpha_2$

18:            $w^i \leftarrow w_2^i$

19:        **else**

20:            $\alpha \leftarrow \alpha_1$

21:            $\alpha_1 \leftarrow \alpha_2$

22:            $N_{\text{eff}} \leftarrow N_{\text{eff}_1}$

23:            $N_{\text{eff}_1} \leftarrow N_{\text{eff}_2}$

24:        **end if**

25:        $i \leftarrow i + 1$

26:        **if** $i = maxiter$ **then**

27:            **print** "failed to converge, duplicating all particles"

28:            $w^i \leftarrow \frac{1}{N_{\text{e}}}$

29:        **end if**

30:    **end while**

31: **end if**

---

is similar to EnKF localisation algorithm where the localisation *domain* is based on background correlations (Hamill et al.,

2001).

The detailed k-localisation algorithm is described in Alg. 2, where:

- The **select_k_biggest** method returns for each variable, the domain $d_v$, of up to $k$ observed points (named $p$) that are the most correlated (in absolute value) with $n$, and match the following criteria, which were adjusted in preliminary experiments:

    - in $x_v^i$, there are at least 10% of members defined in both points. As reflectance is not defined when there is no snow, spurious high correlation can be obtained when the computation of correlations is based on a very low number of pairs.

    - $|\mathbf{B}_v(n,p)| > 0.3$. If the absolute correlation is too low, it is likely that there is a poor potential for the distant observation to constrain the ensemble locally. In such a situation, it is better to reject the observation from the local analysis. Negative ensemble correlations can be physically sound, e.g. after a rain-on-snow event between the HS of two points separated by the rain-snow line. In such a situation, an HS observation on either point can hold information on precipitation rates at both locations. At the observed location, the PF will probably select the members with the most appropriate precipitation rates. This sample is likely to perform well at both locations, so it can be used to constrain the unobserved location.

- $d$ is the collection of the domains $d_v$

- **extract_points** extracts $d$ from $y, \widehat{x}^i$ and $\mathbf{R}$.

### 2.3.3   Particle Filter and reflectance observations

Assimilating reflectance with the PF requires some adaptations. In Crocus, TARTES optical scheme (see Sec. 2.2.1) only provides snow reflectance, not all-surface reflectance: no value for the surface reflectance is issued in the absence of snow. Conversely, the weights of the particles are not defined in Eq. 2 if the members are snow-free. These issues were roughly accommodated by setting the reflectances of snow-free members and observations to 0.2 (the value of bare soil broadband albedo in ISBA model) in the PF Eq. 2 (Sec. 2.2.3).

## 3   Evaluation strategy

Our strategy is to assess the performance of the analysis by means of twin experiments, i.e. using synthetic observations (e.g. Reichle and Koster, 2003). The assimilation run is compared to an identical run without assimilation (open-loop run). Synthetic observations are extracted from a model run and assimilated without adding any noise. These observations mimic real observations with a perfect knowledge of the true state. Analysis and open-loop experiments can therefore be compared with this true state anywhere, for any variable. In a first step, this allows us to get rid of the error and bias issues inherent to

**Algorithm 2** k-localisation algorithm

---

**Input:** $\widehat{x}^i$, $y$, $\mathbf{B}$, $\mathbf{R}$, $N_{\text{eff}}^*$

**Output:** $(w_n^i)_{0 < n \leq N_{\text{pts}}}$

1: **for** $n = 1$ to $N_{\text{pts}}$ **do**
2:   $k \leftarrow k_{\max}$ {try to ingest all available observations.}
3:   $cond \leftarrow$ **True**
4:   **while** $cond$ **and** $k > 0$ **do**
5:     **for** $v = 1$ to $N_v$ **do**
6:       $d_v \leftarrow$ **select_k_biggest**$(n, k, \mathbf{B}_v, \widehat{x}_v^i, y)$
7:     **end for**
8:     $y_k, \widehat{x}_k^i, \mathbf{R}_k \leftarrow$ **extract_points**$(y, \widehat{x}^i, \mathbf{R}, d)$
9:     $w_n^i \leftarrow$ **weights**$(\widehat{x}_k^i, y_k, \mathbf{R}_k)$ (Eq. 2)
10:    $N_{\text{eff}} \leftarrow$ **eff_weights**$(w_n^i)$ (Eq. 3)
11:    **if** $N_{\text{eff}} \geq N_{\text{eff}}^*$ **then**
12:      $cond \leftarrow$ **False**
13:    **end if**
14:    $k \leftarrow k - 1$
15:  **end while**
16:  **if** $k = 1$ **then**
17:    $w_n^i \leftarrow$ **inflation**$(\widehat{x}_k^i, y_k, \mathbf{R}_k, N_{\text{eff}}^*)$
18:  **end if**
19: **end for**

---

real observations (Cluzet et al., 2020), a reason why we did not add any noise to the synthetic observations as commonly done in twin experiments (Lahoz and Menard, 2010). This way, we can focus on the two following questions (see Sec. 1):

- Is CrocO PF able to efficiently spread the information from sparse observations into space without degenerating?

- Is spatial information content of reflectance a valuable source of information for snowpack models?

In order to disentangle these questions, we run baseline experiments assimilating synthetic observations of HS which is strongly linked with SWE (Margulis et al., 2019). These experiments are used to evaluate the PF algorithms efficiency, and as a baseline for synthetic reflectance assimilation experiments evaluating the information content of reflectance.

Three different algorithms are evaluated: the *global* algorithm (with inflation), the *rlocal* algorithm (with inflation) and the k-localized algorithm *klocal*.

## 3.1 Experiments

### 3.1.1 Twin experiments setup

In our twin experiment setup, an open-loop ensemble is used as a reference and to generate synthetic observations. open-loop simulations are carried out with CrocO for 4 consecutive winters (2013-2017) in the Grandes-Rousses (see Sec. 2.1), with 160 members. For each year, the average of SWE over time and space is computed from each member, and members corresponding to the 20th, 40th, 60th and 80th percentiles of the ensemble are extracted to be used as synthetic observations (denoted *year*_p*percentile* e.g. 2014_p80). This method enables us to evaluate the efficiency of data assimilation experiments under contrasted snow condition scenarios. Before any assimilation experiment, the open-loop member ($F_i - M_i$ couple in Fig. 2) used as true state is withdrawn and replaced by a new random member.

The spatial coverage of synthetic observations was reduced, mimicking a typical reflectance mask. Synthetic observations were only available above an assumed constant tree line at 1800 m (see Fig. 1), and not available in steep slopes (over 20º) and in northern aspects (shadows, considering a daily satellite pass around 10-11:00 UTC.), for the whole snow season. As a result, in this case, only 35 (over 187) topographic classes are observed. Observation dates were chosen corresponding to clear-sky days with a MODIS overpass, resulting in an approximately weekly frequency (e.g. Revuelto et al., 2018; Cluzet et al., 2020). Reflectance is sensitive to the surface SSA and LAP (see Sec. 1). A minimal set of two different bands is used, corresponding to MODIS sensor band 4 (555 nm, sensitive to SSA and LAP) and 5 (1240 nm, usually only sensitive to SSA) (e.g. Fig. 2. of Cluzet et al., 2020). Observation error variances are set to $1.0 \times 10^{-2} \mathrm{m}^2$ for HS and $5.6 \times 10^{-4}$ and $2.0 \times 10^{-3}$ for band 4 and band 5 reflectance respectively (Wright et al., 2014). These values are only initial values for the inflation in the *global* and *rlocal* algorithms. Since the *klocal* algorithm only used inflation if $k$ drops to 1 (see Sec. 2), observation error variances are multiplied by a factor of 5 to enable the *klocal* algorithm to ingest observations from several points.

In order to study the ability of the *global*, *klocal* and *rlocal* algorithms to spread the information in space, a first set of experiments is conducted assimilating HS with 40 members (see setup in Tab. 1). In order to evaluate the algorithms ability to assimilate reflectance (Band 4 and Band 5) a second set of experiments is conducted, other things being equal (Tab. 2). The ensemble size is increased from 40 to 160 in a third set of experiments assimilating reflectance, in order to analyse the influence of a larger ensemble on the algorithms performance (Tab. 3). Note in Tab. 1-3 that $N_{\mathrm{eff}}^*$ is adjusted to the ensemble size, in order to preserve $N_{\mathrm{e}}/N_{\mathrm{eff}}^* \approx 5 - 7$ following Larue et al. (2018).

## 3.2 Evaluation Scores

The performance of the assimilation and open-loop run is evaluated against the synthetic truth using several scores. The Absolute Error of the ensemble mean (AEM) and ensemble spread $\sigma$ are two common metrics of ensemble modelling. Given an ensemble $E_{m,c,t}$ of $N_{\mathrm{e}}$ members $m$ in topographic class $c$ at time $t$ and the corresponding truth $\tau_{c,t}$, the ensemble mean is

described by Eq. 4:

$$\overline{E}_{c,t} = \frac{1}{N_{\mathrm{e}}} \sum_{m=1}^{N_{\mathrm{e}}} E_{m,c,t} \tag{4}$$

From which we can compute the absolute error AEM (Eq. 5) and the spread (or dispersion) $\sigma$ (Eq. 6):

$$AEM_{c,t} = |\overline{E}_{c,t} - \tau_{c,t}| \quad \forall (c,t) \in [1, N_{\mathrm{pts}}] \times [1, N_t] \tag{5}$$

$$\sigma_{c,t} = \sqrt{\frac{1}{N_{\mathrm{e}}} \sum_{m=1}^{N_{\mathrm{e}}} (E_{m,c,t} - \overline{E}_{c,t})^2} \quad \forall (c,t) \in [1, N_{\mathrm{pts}}] \times [1, N_t] \tag{6}$$

Where $N_t$ is the number of evaluation time steps.

The Continuous Ranked Probability Score (CRPS, (Eq. 7) Matheson and Winkler, 1976) evaluates the reliability and resolution of an ensemble based on a verification dataset. An ensemble is reliable when events are forecast with the right probability, and has a good resolution when it is able to discriminate distinct observed events. For a reliable system, the resolution is equivalent to the sharpness, which is the spread of the produced forecasts.

If we denote $F_{c,t}$ the Cumulative Distribution Function (CDF) and $\mathcal{T}_{c,t}$ the corresponding truth CDF (Heaviside function centred on the truth value), the CRPS is computed at $(c,t)$ following:

$$\mathrm{CRPS}_{c,t} = \int_{\mathbb{R}} (F_{c,t}(x) - \mathcal{T}_{c,t}(x))^2 dx \quad \forall (c,t) \in [1, N_{\mathrm{pts}}] \times [1, N_t] \tag{7}$$

In this work, $\mathrm{CRPS}_{c,t}$ value is averaged over time alone or time and space depending on the desired level of aggregation.

The CRPS can be decomposed in two terms following Candille et al. (2015):

$$\mathrm{CRPS} = \mathrm{Reli} + \mathrm{Resol} \tag{8}$$

Where Reli quantifies the reliability of the ensemble. The associated skill scores (CRPSS and ReliS) can be used to compare the performance of an ensemble $E$ to a reference $R$, here, the open-loop run:

$$\mathrm{CRPSS}(E) = 1 - \frac{\mathrm{CRPS}(E)}{\mathrm{CRPS}(R)} \tag{9}$$

A skill score of 1 denotes a perfect score, 0 a neutral performance and $-\infty$ is the worst achievable skill score.

## 4 Results

### 4.1 Preliminary Results

#### 4.1.1 Impact of the inflation

The inflation algorithm was introduced by Larue et al. (2018) in point scale simulations but to the best of our knowledge, never applied in a spatialised context. Here we evaluate its impact on the *global* algorithm by switching it on/off. As an example,

Fig. 3 shows the impact of the inflation on SWE when assimilating the HS of 2015_p80 (as defined in Sec. 3.1.1) member with the *global* algorithm, in a topographic class which is not observed (1800_N_40, as defined in Sec. 2.1). This choice of member and topographic class is representative of the impact of the inflation on the *global* algorithm.

In this case, both inflation ($N_{\text{eff}}^* = 7$) and no inflation ($N_{\text{eff}}^* = 1$) lead to a significant reduction of the ensemble spread compared with the open-loop (Fig 3b). From January 2015 until the peak of SWE in mid-April 2015, (Fig. 1c) the simulation with inflation has significantly lower errors than without inflation and the open-loop (10-20 $\text{kg m}^{-2}$ vs. 60-80 $\text{kg m}^{-2}$ and 30-50 $\text{kg m}^{-2}$ respectively), leading to a better agreement with the synthetic truth in the melting season (Fig. 3a). During the melting season (mid-April 2015 onwards), the AEM of the assimilation algorithms is reaching a peak, coinciding with an absence of

observations. In comparison, the open-loop AEM is smaller in the first part of the melting season, but the spread is three times larger, making it almost uninformative. For several analyses (2014, November 21[th], and 2014, December 30th for example) the ensemble spread without inflation drops to 0 while its AEM strongly increases compared to the open-loop, suggesting that it is prone to degeneracy.

### 4.1.2   Correlation patterns

The *klocal* algorithm relies on background correlation patterns to define localisation *domains*. To illustrate the potential of using such information in the PF, Fig. 4 shows the correlation patterns of the 40 members open-loop in a unobserved topographic class (1800_N_40, red dot) in the mid-winter (2015, February 20[th]), several months after the snow season onset. The assimilation variables exhibit strong but contrasting correlation patterns. Band 4 (Fig. 4a) correlations are generally high (0.6-

330 1) and uniform. Many of the observed classes (black dots) are strongly correlated with the considered class. Similar results are obtained for HS (Fig. 4c). Band 5 (Fig. 4b) exhibits substantial correlations, in particular across slopes. However, they are more restricted to the northern aspects, only a few observed classes in the Eastern aspects being substantially correlated with the considered class. Note that negative correlations are evidenced with some lower altitude South-oriented topographic classes (e.g. 1500_S_40 on Fig. 4b). Finally, these patterns vary with time but remain substantial along the whole season (not shown),

and increasing the ensemble size up to 160 leads to identical patterns (not shown).

## 4.2   Results of the experiments

### 4.2.1   Assimilation of the Height of Snow

In a first step, assimilation of HS from the different synthetic observation scenarios was conducted to serve as a reference for

reflectance assimilation. Fig. 5 shows the CRPSS (Eq. 9, aggregated over time only) of the HS assimilation with the three PF algorithms considering the synthetic member 2013_q20 as reference. Results for this specific synthetic member were chosen here as a representative example of the algorithms performance.

The *rlocal* performance compared with the open-loop is high (0.7-1), but limited to the observed classes (black dots) since there

is no spatial propagation in this algorithm. *global* and *klocal* algorithms have similar, overall good performance, managing to strongly reduce modelling uncertainties except at very low altitudes (600-900 m), (skills of -0.2) where snow does not usually last for more than a few weeks.

This behaviour may vary with the snow conditions, i.e., between the different assimilated synthetic observation scenarios and from one year to another. In order to generalize this result, Fig. 6 shows the CRPS and Reli (aggregated over time and space) of the different algorithms for the 16 synthetic observation scenarios and differentiated between observed and unobserved classes. CRPS and Reliability are considerably reduced compared with the open-loop (by a factor of 2-3 and 4-5, respectively) for all the algorithms in the observed classes. This suggests that the PF manages to reduce the spread of the ensemble while reducing its errors. In the unobserved classes, the gain is almost as good (CRPSS of 0.6) except for the *rlocal* algorithm, which is identical to the open-loop as expected. No significant difference of skill is obtained between *global* and *klocal* algorithms.

### 4.2.2   Assimilation of Reflectance

Optical reflectance is a promising assimilation variable due to its extended availability in satellite observations, but assimilation of raw reflectance products is not expected to constrain bulk variables like SWE or HS as much as HS assimilation. In order to assess this difference, we conduct assimilation of reflectance only, in the same setup as in Sec 4.2.1, all other things being equal.

Fig. 7 shows the performance of the reflectance assimilation for the 16 synthetic observation scenarios with 40 members (filled boxes). The different algorithms only lead to moderate improvements in CRPS (median CRPSS of 0.-0.2, median ReliS of 0.2-0.4). Moreover, the *global* and *klocal* algorithms frequently degrade the performance, suggesting that this configuration is not robust.

Suspecting that 40 members is insufficient to properly represent the multivariate probability density function of reflectance and other model variables, the ensemble size was increased to 160 (hatched boxes), leading to marked improvements in the performance and robustness of the algorithms (median CRPSS of 0.2, median Reli of 0.4-0.6). Reliability of the *global* algorithm is significantly improved compared to the *klocal* algorithm.

Fig. 8 shows the spatial performance of the different algorithms for member 2016_p60. Spatial patterns similar to the HS assimilation are found. *rlocal* performance is limited to the observed classes, while *global* and *klocal* manage to improve the simulations across aspects and slopes. However, skill scores are lower than for HS (0.2-0.5), and the performance of all algorithms is poor in the classes that are the farther away from the observations, i.e. at lower elevations (600-900 m) and in some of the high altitude steep Northern classes (e.g. 2100_N_40 on Figs. 8b-c). Finally note that slight degradations of performance can sometimes be evidenced even in the observed classes for all the algorithms (e.g. in flat conditions at 3300 m on Fig. 8a for the *rlocal*, not evidenced by this example for the other algorithms).

## 5   Discussion

In this section, we discuss the performance of CrocO PF algorithms using the assimilation of HS, and consider the potential of the assimilation of reflectance in view of assimilating real data.

### 5.1   Tackling Particle Filter degeneracy

Because they assimilate several observations at the same time, *global* and *klocal* approaches could be prone to PF degeneracy. However, they almost never degrade the performances when assimilating HS in a variety of years and synthetic observation scenarios percentiles (Fig. 6). This suggests that either inflating the observation errors (as demonstrated by Larue et al. (2018), a result we have generalized in space) or exploiting background correlations to reduce the number of assimilated observations, are two efficient approaches to tackle degeneracy.

In several cases though, a strong degradation of score occurs when assimilating reflectance (Fig. 7), which could either be attributed to an algorithmic failure in the PF, or an intrinsic lack of informativeness of reflectance in some situations. Based on the good behaviour of the algorithm with HS, and because by construction, *global* and *klocal* algorithms cannot lead to a degenerate PF sample we consider this comes from the reflectance itself (this point will be further discussed in the following sections).

Beyond tackling degeneracy, *global* and *klocal* algorithms also beat the *rlocal* approach on Reli and CRPS scores (Figs. 7 and 8). This suggests that assimilating multiple observations increases the quality of the PF analysis, even locally. More precisely, most of the improvement is due to the Reli term of the CRPS. This property is crucial for ensemble modelling, because it ensures that events are forecasted with a right frequency. However, this is not sufficient, e.g. the climatology has a perfect reliability but is not informative at all. Successful assimilation manages to improve general metrics such as the CRPS while improving the reliability. On this aspect, the *global* and *klocal* algorithms have a satisfying performance.

### 5.2   Propagating the observations information

Having sparse observations is one of the most challenging tasks for data assimilation systems of snowpack observations (Magnusson et al., 2014; Largeron et al., 2020). In our partially observed, synthetic setup, the *global* and *klocal* PF variants developed here efficiently propagate the observations information to the unobserved classes, with generally a better performance than the open-loop and the *rlocal* approach in the unobserved classes when assimilating HS (Fig. 5).

The algorithms performance is particularly good across aspects and slopes with only a few steep, northern aspect slopes exhibiting neutral to poor performances (Figs. 5 and 8). This suggests that southern aspect and flat classes are informative for the majority of the simulation domain. Conversely, considering that there are strong background correlations between the western and eastern sides of the domain, we can speculate that observing either side could yield overall good results.

On these figures, propagation of the information is limited towards lower elevation (600-1200 m). At such elevations, the snow cover is usually intermittent and a good discrimination of the precipitation phase is crucial. The PF does this indirectly

through HS and reflectance observations, because rain causes a decrease of HS through compaction and melting while Band 4 and Band 5 reflectances also decreases because of quick isothermal metamorphism (i.e. the surface SSA decreases). However, in our setup, the lowest observed elevation is 1800 m, therefore indirect observation of the rain-snow line positioning under this level is not possible, potentially explaining the moderate performance of the PF there. In that case, assimilation of Snow Cover Fraction might be the best solution: since the snowpack is intermittent there, the informativeness of this variable is maximal

(Aalstad et al., 2018).

*Global* and *klocal* algorithms exhibit strong performances when assimilating HS (Fig. 5). HS is closely linked with the SWE (by the bulk density) and the interest of this variable for data assimilation is clear (Margulis et al., 2019). Here, it should be kept in mind that HS assimilation is used as a baseline experiment to evaluate the algorithms and put reflectance assimilation into perspective. The prescribed HS observation errors ($\sigma_0 = 0.1$m) are not necessarily realistic. They should be adapted to

the nature of the HS sensor. For example, space-borne HS observation errors are typically larger (e.g. Eberhard et al., 2020; Deschamps-Berger et al., 2020). The assimilation of such observations would probably yield lower improvements.

Though the performance is lower for Reflectance than in our HS experiments, it remains considerable and in line with previous results on point simulations (Charrois et al., 2016), with an average score improvement of 20-40%. This study quite surprisingly suggests that reflectance information can be spread from southern slopes to the northern ones, although in many

situations, the snowpack evolves in different ways for these two aspects. For example, in sunny conditions, melt and wet metamorphism will cause a drop in reflectance in southern slopes, while reflectance will not evolve much in northern slopes. Such a phenomenon could explain why low background correlations between southern and northern aspects are exhibited in Band 5 (Fig. 4), which is the most sensitive to surface metamorphism through SSA. This example shows that Band 5 reflectance observations in southern slopes are not necessarily informative on Band 5 reflectance values in the northern aspect per se on

every date. On average, however, the positive impact of reflectance observations suggests that they enable the PF to reject the ensemble members with inadequate meteorological forcings (snowfall or cloud cover would lead to wrong reflectance values), or multiphysical parametrisations (influencing e.g. the surface metamorphism), thus correcting the ensemble in the whole domain. These insights are consistent with the study of Winstral et al. (2019), where in situ observations are used to correct meteorological forcing parameters across large simulation domains.


Regarding the observations, our study has some methodological limits, however. Observation errors are very roughly prescribed, and the assimilated observations are not corrupted as usually done in synthetic experiments (e.g. Durand and Margulis, 2006). These choices were motivated by the fact that very little is known about the spatial correlation of reflectance observation errors in the semi-distributed setting (e.g. Cluzet et al., 2020). In a recently submitted paper, the impact of random and sys-

tematic errors of reflectance observations on point-scale assimilation experiments is thoroughly investigated (Revuelto et al., in prep). Efforts to better characterize the spatial structure of these observation errors should be conducted in future work.

## 5.3 Towards the assimilation of real observations of reflectance

Reflectance is an appealing variable for snowpack modelling because of its sensitivity to snowpack surface properties (Dozier
et al., 2009) and the abundance of moderate to high resolution space-borne sensors (MODIS, Sentinel2-3, VIIRS, Landsat...)
providing us with a handful of observations to assimilate, contrary to HS. The potential for assimilation of SCF, which is
retrieved from reflectances, is clear (Margulis et al., 2016; Aalstad et al., 2018; Alonso-Gónzalez et al., 2020). This study
demonstrates the potential of the PF to spread information and assimilate raw reflectances with a positive impact (Sec. 5.2).
Yet, assimilating real observations of reflectance is another challenge, for two reasons.

First, space-borne reflectance observations are generally noisy and biased (e.g. Cluzet et al., 2020). Satellite retrievals could
be improved in the future (Kokhanovsky et al., 2019; Lamare et al., 2020), and Cluzet et al. (2020) showed that assimilating
ratios of reflectance could be a workaround to tackle this issue. In the near-infrared, the signal-to-noise ratio of reflectances
observations might be sufficient to constrain the surface microphysical properties (Durand and Margulis, 2007; Mary et al.,
2013), whereas the required accuracy for visible reflectance retrievals to remain informative on the snowpack light absorbing
particles content is high (Warren, 2013), and it is yet to prove whether either approach can achieve this requirement.

Second, in this twin experiment framework, spatial patterns of the synthetic observations are likely compatible with the ensemble since they come from the same modelling system. This may not be the case in reality, therefore making it more difficult to
assimilate, and we refer to this issue as model or ensemble realism.

We must assess the strengths and weaknesses of the *global* and *klocal* approaches facing those two issues. The *global* algorithm
assumes that a global optimum can be found across the whole domain, e.g. the information from the different observations is
consistent and can be ingested in one block by the PF. With this strategy, the degeneracy due to the size of the observation
vector is efficiently mitigated by the inflation algorithm as discussed in Sec. 5.1. The *klocal* approach considers that only a
fraction of the observation information is relevant to constrain the model state at a given location. This algorithm tries to ingest
as much information as possible while rejecting observations coming from too statistically different snowpack conditions. As
a consequence, because we do not account for the real spatial patterns of observation errors, and because we work in a twin
experiment setup, a global optimum on the whole domain can exist and can be found by the *global* algorithm. This might be
a reason why it beats the *klocal* approach (Figs. 6 and 7). In the real world, from the model point of view, there might be
contradictory information among the observations that would be difficult to disentangle with a *global* strategy. The *klocal* algorithm could be more suited to this situation, because it is looking for local optima, based on the assumption that background
correlations are a realistic representation of modelling errors.

These background correlation structures could be overestimated by the ensemble, and tests with real observations are necessary. Strong Band 4 correlations (Fig. 4a) might be due to the spatially homogeneous perturbations of LAP fluxes used to force
the simulations (see Sec. 2.2.2), a key driver of this variable, and because the same snow model configuration is applied for a
given member across the simulation domain. Several studies suggest that LAP fluxes vary with elevation and other topographic
parameters (de Magalhães et al., 2019; Sabatier et al., 2020), but to date no reliable model of such processes exists in complex
terrain. In such a context, assuming uniform LAP forcing seems a reasonable compromise. Strong and almost uniform HS cor-

relations (Fig. 4b) might be caused by the spatial homogeneity of precipitation perturbations and because we do not account for e.g. wind drift, intra massif heterogeneity of meteorological conditions and gravitational redistribution of snow (Wayand et al., 2018). Despite this semi-distributed framework suffering from obvious limitations, the potential for high-resolution snowpack modelling (Vionnet et al., 2020; Fiddes et al., 2019; Marsh et al., 2020) is hampered by large errors of the NWP models in mountainous areas (e.g. Nousu et al., 2019).

In the future, improving the ability of ensemble correlations to represent modelling errors could make the spreading of information an even more challenging task with the *klocal* algorithm. But there should remain significant potential for information propagation, as suggested by results at larger scales (Magnusson et al., 2014; Cantet et al., 2019). The potential de-correlation of topographic classes would also impact the *global* algorithm. In a unobserved class, constraining the state of the snowpack with information from areas that are not linked with it would likely degrade the forecasting skill, as suggested by the poor performance of the algorithms at low altitudes (Figs. 5 and 8). On the contrary, applying CrocO over larger domains (e.g. distributed simulations or a collection of semi-distributed *massifs*), would probably see the *klocal* algorithm outperform the *global*. The increased domain size would make it less plausible to find a global optimum over the domain, whereas spatial flexibility would be an asset of the *klocal* algorithm. Finally, in the case of modelled coupling between simulation points (e.g. snow drift), which was not the case here, the spatial discontinuities of the *klocal* analyses (see Sec. 1) might be a drawback compared to the global approach. Spatial discontinuities may reveal impractical for the interpretation of individual simulations outputs by snow forecasters too. The *klocal* approach is likely to reduce these discontinuities compared to the *rlocal*, because similar locations will receive similar analyses (i.e. based on similar sets of observations). This issue could be partly mitigated by e.g. *state-block-domain* approaches (Farchi and Bocquet, 2018).

### 5.4 Outlook for ensemble modelling and data assimilation

In the snowpack modelling community, ensemble modelling appears as a powerful tool to represent modelling uncertainties (Vernay et al., 2015; Richter et al., 2020) and for data assimilation (Essery et al., 2013; Lafaysse et al., 2017; Piazzi et al., 2018; Aalstad et al., 2018). This study offers a novel approach to extract valuable information on the snowpack spatial behaviour from spatial correlation patterns of the ensemble. These patterns could be used to diagnose links between locations, transfer information between areas, or assess the representativeness of point simulations. More broadly, ensemble background correlations have been exploited for long in the NWP and oceanographic communities to refine modelling errors representation which led to significant improvements in the DA systems (Evensen, 2003; Buehner, 2005).

Ensembles might open a way for the assimilation of point scale observations, or sparse remotely-sensed observations into spatialised simulations of the snowpack as suggested by Winstral et al. (2019) and the present work. For instance, there are numerous snow gauges and snow pit observations in the ski resorts of the French Alps. These data could be assimilated to correct the ensemble in spatialised simulations (Winstral et al., 2019). The spatial pattern of assimilated observations in the experiments of Sec. 4 do not correspond to the real-life spatial coverage of this kind of observations. To give an insight of their

potential, we also applied our methodology to assimilate only five synthetic HS observations with the *global* PF in the 1200 m to 2400 m flat classes. The results are shown in Fig. 9. The assimilation improves the performance in all aspects and slopes. Naturally, this suffers from the same limitation as discussed in Sec. 5.3, not to mention the limited spatial representativeness of in situ observations but it shows some potential for this idea.

In that way, a more rational use of the available observations could be implemented towards a new ensemble data assimilation system. In the present CrocO system, SAFRAN reanalysis are only assimilating weather station information (precipitation phase, temperature, wind), and makes no use of the numerous snow observations available. Here, snow observations are assimilated by the PF, but are not used to correct meteorological forcings (only snow variables, see Fig. 2). In a new ensemble data assimilation system, within CrocO, the SAFRAN meteorological analysis could be bypassed, the PF operating directly both on the meteorological and snowpack variables in a more comprehensive and coupled strategy.

## 6   Conclusions

In this study, we introduced CrocO, a new ensemble data assimilation system able to reduce the errors of a spatialised snowpack model in locations that are not observed. The ensemble is built by a combination of meteorological and multi-physical ensembles to represent modelling uncertainties. A Particle Filter assimilates observations of HS and Reflectance. We developed two variants of the PF using inflation or k-localisation, in order to spread the information from partial observations of the system, without degeneracy of the PF. In the framework of synthetic experiments, we have shown in particular that:

1. These variants are able to ingest numerous observations without degeneracy;

2. An efficient spreading of the observations information towards the unobserved areas is achieved with the *global* and *klocal* approaches;

3. Reflectance assimilation leads to an overall 20% improvement in CRPS and 60% in reliability.

We suggest that this approach could be used in any spatialised framework to assimilate sparse observations from e.g. networks of in-situ snowpack observations. Beyond the snowpack modelling community, the inflation and k-localisation strategies could help address the problem of partially observed systems. This work is also a first step towards the operational assimilation of reflectance in a semi-distributed context. To reach that goal, biases of reflectance retrievals should be studied, and observation error structures duly quantified. Snow cover fraction would be a good companion variable to jointly assimilate with reflectances, requiring the use of an appropriate observation operator. Extending the simulation domain to several massifs would allow the exchange of information between neighbouring massifs with the *klocal* algorithm.

*Code availability.*   The Crocus snowpack model (including all physical options of the ESCROC system) and the Particle Filter algorithm are developed inside the opensource SURFEX project. The source files of SURFEX code are provided at https://doi.org/10.5281/zenodo.3774861

to guarantee the permanent reproductibility of results. However, we recommend potential future users and developers to access to the code from its git repository (git.umr-cnrm.fr/git/Surfex_Git2.git) to benefit from all tools of code management (history management, bug fixes, documentation, interface for technical support, etc.). This needs a quick registration, the procedure is described at https://opensource.cnrm-game-meteo.fr/projects/snowtools/wiki/Procedure_for_new_users. The version used in this work is tagged as CrocO_v1.0.

A python software called CrocO_toolbox was specifically developed, in order to pre-post process and launch CrocO experiments. It is available on Github (https://github.com/bertrandcz/CrocO, release v1.0) along with a documentation.

The article version of CrocO_toolbox is archived at: https://doi.org/10.5281/zenodo.3784980. This software strongly relies on two external python projects ensuring the files management between the different steps of a simulation and the interface with Meteo-France HPC system

(including parallelization and data storage): snowtools and vortex. Their sources are available at https://doi.org/10.5281/zenodo.3774861 (same archive as SURFEX) to guarantee the permanent reproducibility of results. However, as for the SURFEX project and for the same reasons, it is recommended to access snowtools code from its git repository (git.umr-cnrm.fr/git/snowtools_git.git). The version used in this work is also tagged as CrocO_v1.0. The vortex project gathers all environment-specific codes of Météo-France modelling systems relative to its HPC computing system. For this project, only the sources which are specific to this article simulations are provided. The common objects

inheritance is based on Vortex version 1.6.1. The version used in this work is also tagged as CrocO_v1.0 in the vortex git repository.

Because these software could not be applied outside Météo-France HPC environment, CrocO python software offers the possibility to run CrocO simulations locally. This functionality was not used here due to the high numerical cost of our simulations, which required the use of Météo-France HPC environment.

*Data availability.* Input and output data necessary to reproduce the manuscript simulations and figures are provided at https://doi.org/10.5281/zenodo.3775007. This archive includes : SAFRAN reanalyses, (also available at https://doi.org/10.25326/37), MOCAGE forcings, namelists, configuration files and spinup files necessary to reproduce the simulations. Raw model outputs can be provided on request but since they amount up to 500+ Gigabytes, only post-processed simulations outputs are provided in this archive, along with scores and scripts to reproduce the manuscript figures.

**Appendix A: Stochastic perturbations of the forcings**

The stochastic perturbation procedure of the forcings is introduced in Sec. 2.2.2 and is identical to Charrois et al. (2016) for the meteorological parameters and Cluzet et al. (2020) for the light absorbing particles (LAP) fluxes. For a given date and forcing variable, perturbation values are the same for all the points in space (no spatial auto-correlation is considered), as SAFRAN semi-distributed massifs have a limited spatial extent (about $1000 \ \mathrm{km}^2$). Precipitation, incoming radiations, wind speed and air

temperature from SAFRAN are perturbed with temporally autocorrelated stochastic parameters. The precipitation, incoming shortwave radiation, and wind speed are perturbed with a multiplicative noise. Longwave radiation and air temperature are perturbed with an additive noise.

For meteorological variables, the perturbation vector $V$ is built as follows:

$$V(t) = \phi V(t-1) + \varepsilon(t) \tag{A1}$$

Where $\phi = e^{-dt/\tau}$, with $dt$ the forcing timestep, $\tau$ the decorrelation time (in h) and $\varepsilon$ a normal law of mean 0 and variance $\sigma^2(1-\phi^2)$. Parameter values for each variable are described in Tab. A1. The significantly high auto-correlation time of precipitation 1500 h was tuned to roughly adjust the ensemble spread to the observed intra-massif variability of yearly-cumulated precipitation. Note that the precipitation phase is adjusted with the perturbed air temperature to ensure a physical consistency. Further details on the procedure can be found in Charrois et al. (2016).

Regarding LAP fluxes, dry and wet black carbon and mineral dust deposition fluxes from MOCAGE are perturbed with a random factor which keeps constant throughout the year. Each member has a single multiplicative factor following a log-normal law of mean $\mu$ and variance $\sigma$ (see Tab. A2). The mean of black carbon random perturbations was adjusted based on comparisons between simulations and field observations at col du Lautaret, a mountain pass within the considered SAFRAN massif.

## Appendix B: Complements on the implementation

### B1 Technical implementation and code performance

CrocO is implemented within Météo-France HPC (High Performance Computing) environment, enabling to fully parallelize the ensemble (one core per member), and bridge the gap with operational applications (Lafaysse et al., 2013; Morin et al., 2020). This implementation is strongly parallel. As an example, the execution time of a one-year assimilation run of 187 model points with 160 members on 4 nodes of 40 cores each lasts for only two hours. The PF is a lightweight algorithm, most of the computational burden owing to the propagation of the ensemble and input/output. Note also that no significant difference in execution time can be noted between the different PF algorithms.

### B2 PF sample reordering

As mentioned in Sec. 2.3, a reordering step was implemented after the PF resampling from Kitagawa (1996), for practical reasons.

- (3) from $s$, build $\tilde{s}$ such that all elements of the unique values of $s$ lie in the position given by their value. Example with 16 particles:

$$s = [1,1,2,3,3,3,8,8,9,9,9,9,9,16,16,16] \Rightarrow \tilde{s} = [1,2,3,1,3,3,8,8,9,9,9,9,9,16,16,16]$$

Indeed, I/O represents a bottleneck in the PF. When building the analysis $\widehat{X^a}$, the background $\widehat{X^b}$ is already loaded in memory. Since $\widehat{X^a}$ is just a reordering of $\widehat{X^b}$ columns based on $s$, a reordering of $s$ avoids to build a copy of $\widehat{X^b}$. This way, $\widehat{X^a}$ is built

by an online modification of $\widehat{\mathbf{X}^\mathrm{b}}$ using two pointers. Reordering is a growing consideration in the PF community (Farchi and Bocquet, 2018).

*Author contributions.* BC wrote the manuscript, BC, ML and MD designed the study, BC developed the code with help from ML, L-FM and CA. BC, MD, ML and EC designed the PF variants. All authors contributed to results analysis and discussion.

*Competing interests.* The authors have no competing interests to declare.

*Acknowledgements.* The authors are grateful to François Tuzet, Jesus Revuelto, Rafife Nheili, César Deschamps-Berger, Stéphanie Faroux (SODA) and Matthieu Vernay for their precious help in collecting input data and code implementation. They also would like to thank Fanny Larue, Joseph Bellier, Pierre De Mey and Guilhem Candille for helpful discussions on the PF inflation and CRPS decomposition.

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

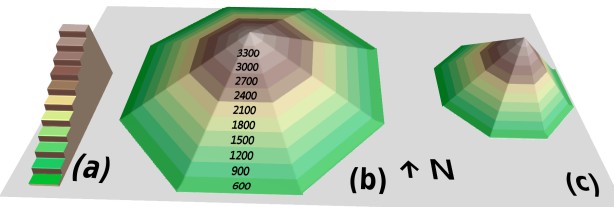

**Figure 1.** 3D schematic view of the semi-distributed geometry, where the numbers represent the altitudes of the elevation bands (in m). From left to right, the three different mountains represent the flat, 20° and 40° slopes.

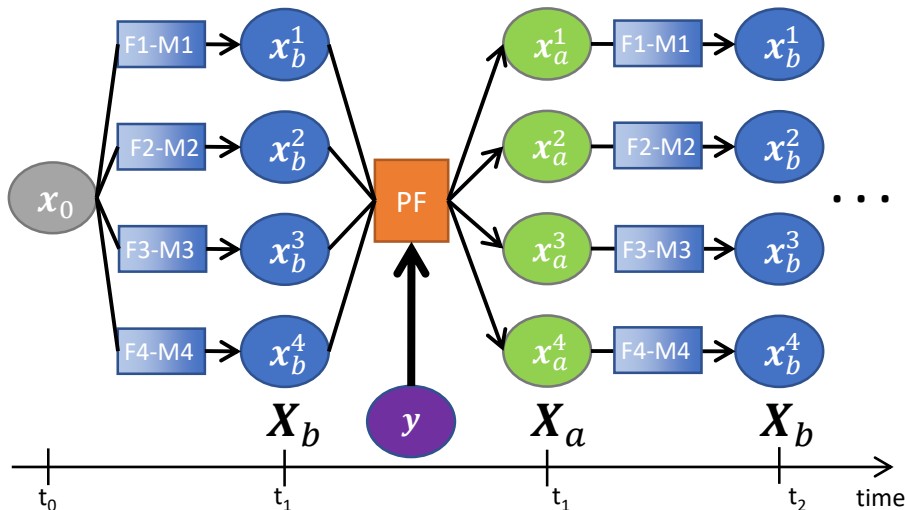

**Figure 2.** Workflow of CrocO ensemble data assimilation system with 4 members. $\widehat{x_0}$: initial state at time $t_0$, Fi: forcing, Mi: ESCROC member, $\widehat{X_b}$: background state, $\widehat{x_b^i}$: background particles, $\widehat{X_a}$: analysis, $\widehat{x_a^i}$: analysis particles, $y$: observation, $t_1$ and $t_2$: observation dates.

| PF Algorithm | $N_\mathrm{e}$ | inflation | $N^*_\mathrm{eff}$ | HS $\sigma_o^2$ (m$^2$) |
|---|---|---|---|---|
| rlocal | 40 | on | 7 | $1.0 \times 10^{-2}$ |
| global | 40 | on | 7 | $1.0 \times 10^{-2}$ |
| klocal | 40 | on (if k=1) | 7 | $5.0 \times 10^{-2}$ |

**Table 1.** Setup for the height of snow assimilation experiment.

| PF Algorithm | $N_e$ | inflation | $N_{eff}^*$ | B4 $\sigma_o^2$ | B5 $\sigma_o^2$ |
|---|---|---|---|---|---|
| rlocal | 40 | on | 7 | $5.6 \times 10^{-4}$ | $2.0 \times 10^{-3}$ |
| global | 40 | on | 7 | $5.6 \times 10^{-4}$ | $2.0 \times 10^{-3}$ |
| klocal | 40 | on (if k=1) | 7 | $2.8 \times 10^{-3}$ | $1.0 \times 10^{-2}$ |

**Table 2.** setup for the first reflectance assimilation experiment.

| PF Algorithm | $N_e$ | inflation | $N_{eff}^*$ | B4 $\sigma_o^2$ | B5 $\sigma_o^2$ |
|:---:|:---:|:---:|:---:|:---:|:---:|
| rlocal | 160 | on | 25 | $5.6 \times 10^{-4}$ | $2.0 \times 10^{-3}$ |
| global | 160 | on | 25 | $5.6 \times 10^{-4}$ | $2.0 \times 10^{-3}$ |
| klocal | 160 | on (if k=1) | 25 | $2.8 \times 10^{-3}$ | $1.0 \times 10^{-2}$ |

**Table 3.** Setup for the second reflectance assimilation experiment.

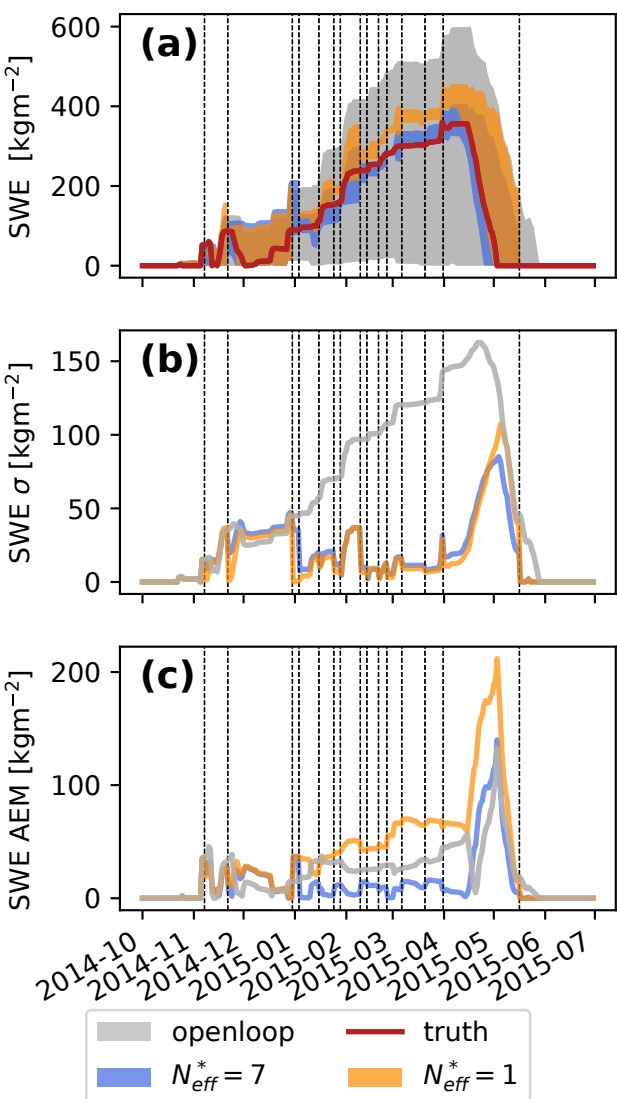

**Figure 3.** Impact of the inflation ($N_{\text{eff}}^* = 7$) versus no inflation ($N_{\text{eff}}^* = 1$) in the 1800_N_40 topographic class (not observed), when assimilating HS of 2015_q80 with the *global* PF. (a) SWE minimum-maximum envelopes as a function of time, (b) spread and (c) AEM. Dashed lines represent the assimilation dates.

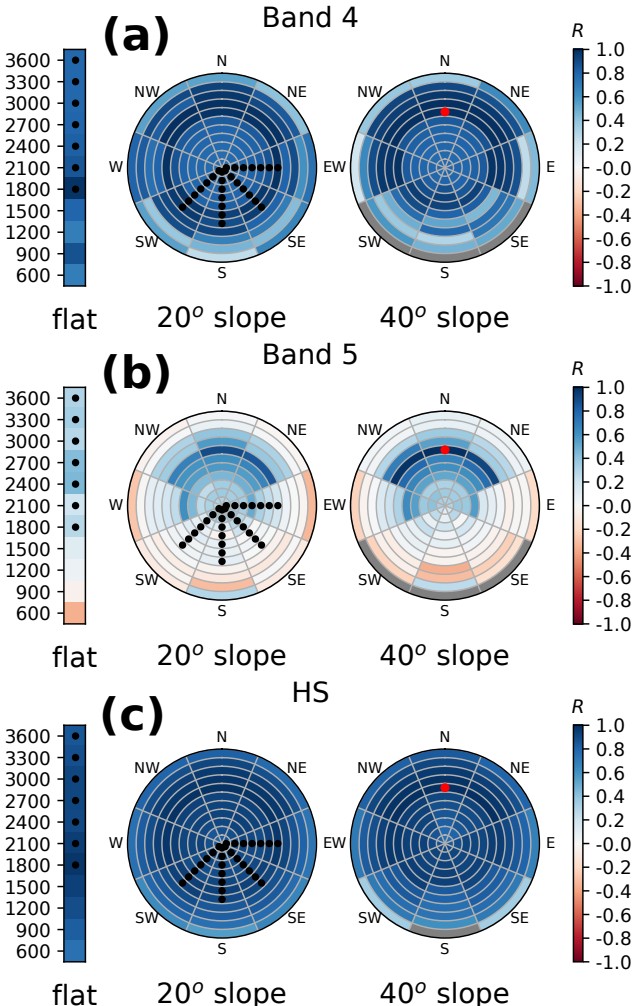

**Figure 4.** 2015, February 20[th] open-loop (40 members) Pearson correlations between the domain points and the 1800_N_40 topographic class (red dot), in Band 4 (a), Band 5 (b) and HS (c). Left bars show the flat topographic classes in the associated elevation bands, while pie plots show the 20° and 40° slope topographic classes, as depicted in Fig. 1. Black dots denote the observed classes.

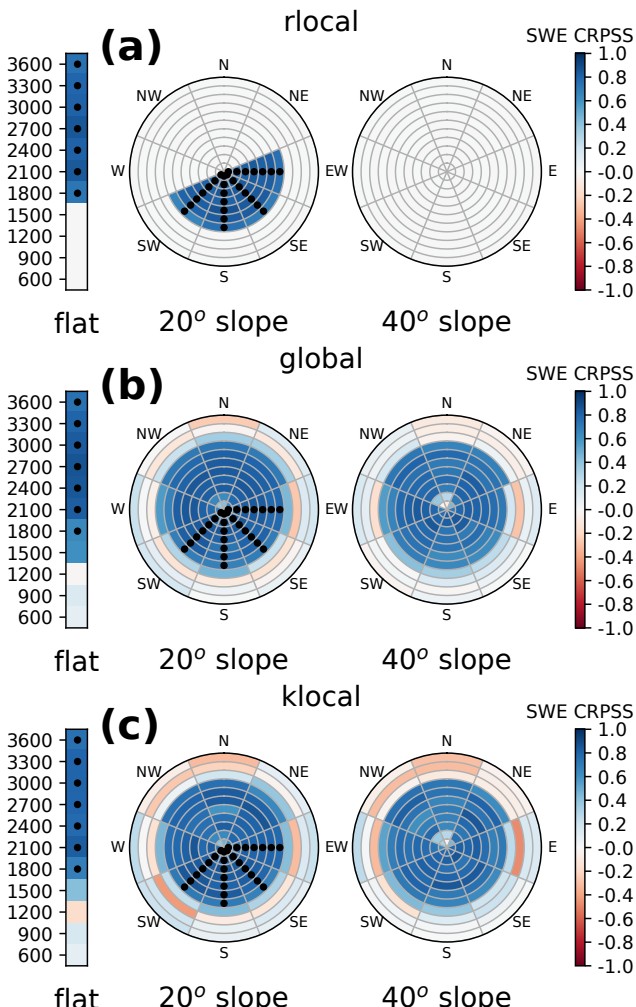

**Figure 5.** CRPS skill score of SWE for the *rlocal* (a), *global* (b) and *klocal* (c) algorithms assimilating the HS of 2013_p20 synthetic observation scenario. The score is computed for the whole snow season for each topographic class. Black dots denote the observed classes.

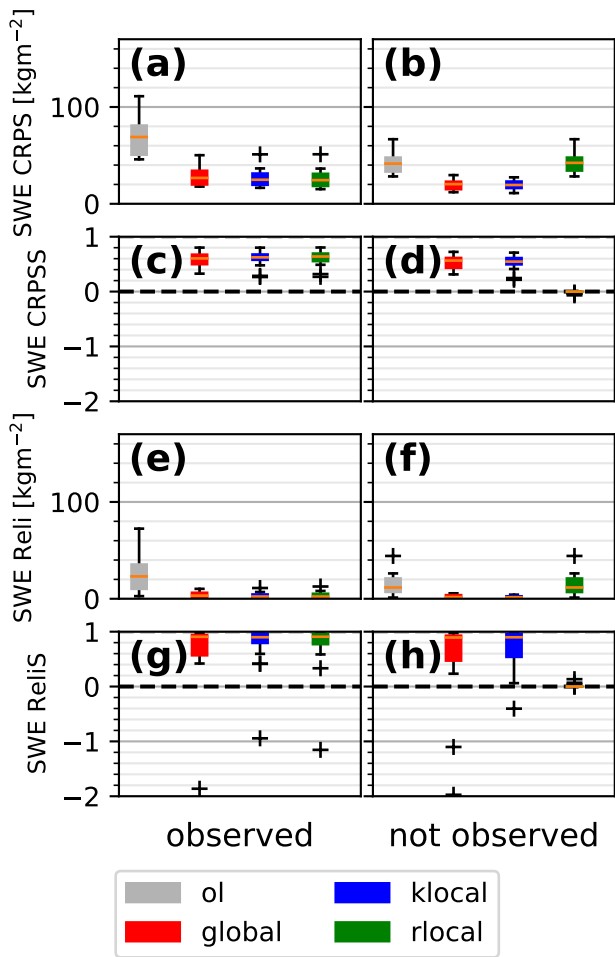

**Figure 6.** Boxplots of SWE CRPS (a,b) and Reli (e,f) for the different algorithms for the 16 different synthetic observation scenarios, separated between observed (left column) and not observed (right panel) classes. Panels (c,d) and (g,h) show the associated skill scores.

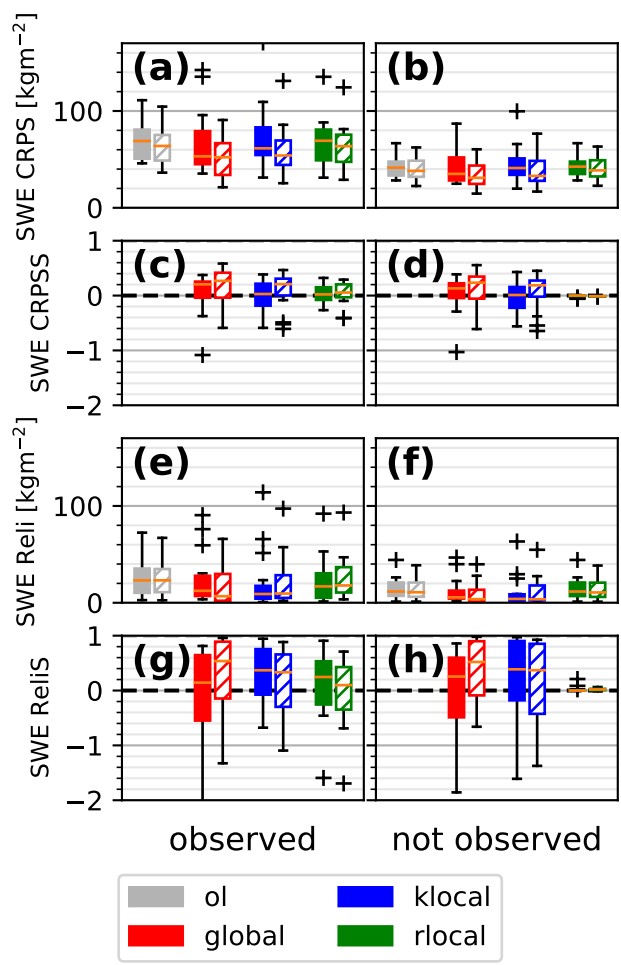

**Figure 7.** Same as Fig. 6 for reflectance with 40 members (filled) and 160 members (hatched).

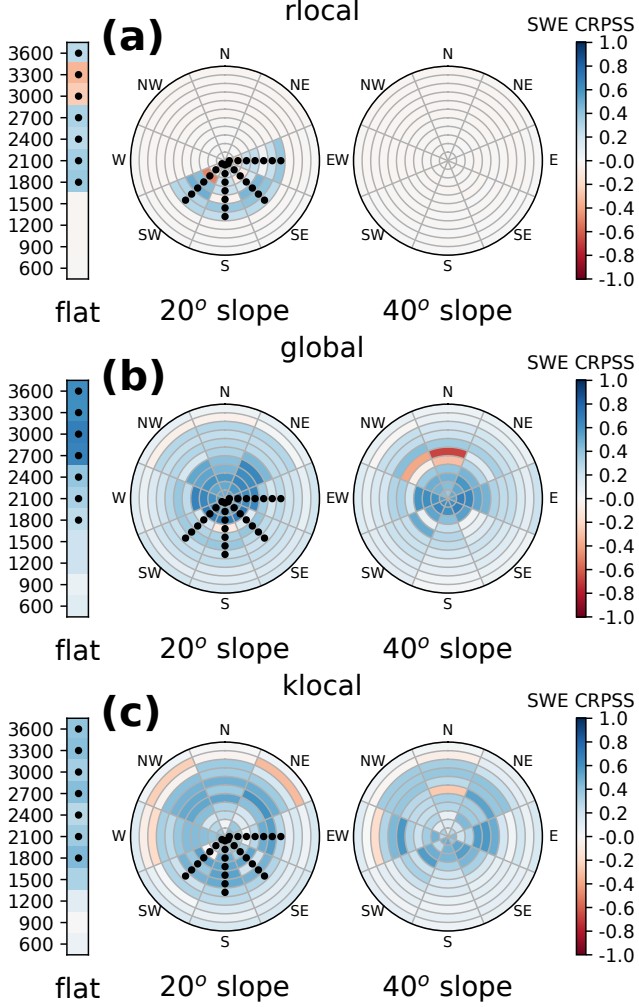

**Figure 8.** Same as Fig. 5 for the assimilation of the reflectance of 2016_p60 synthetic observation scenario.

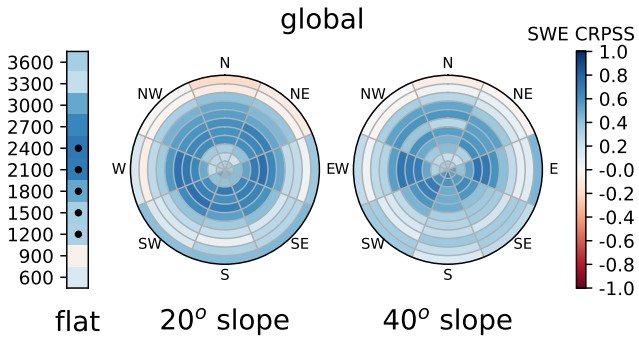

**Figure 9.** Same as Fig. 5 for the assimilation of HS of 2016_p60 synthtic observation scenario in the 1200-2400 m flat classes.

| Variable | Perturbation | $\sigma$ | $\tau$ (h) |
|---|---|---|---|
| Precipitation $(\mathrm{kgm^{-2}h^{-1}})$ | Multiplicative | 0.7 | 1500 |
| Shortwave radiation $(\mathrm{Wm^{-2}})$ | Multiplicative | 0.7 | 3 |
| Wind speed $(\mathrm{unitms^{-1}})$ | Multiplicative | 0.6 | 100 |
| Longwave radiation $(\mathrm{Wm^{-2}})$ | Additive | 24.5 $\mathrm{Wm^{-2}}$ | 30 |
| Air Temperature (K) | Additive | 1.08 K | 15 |

**Table A1.** Perturbation parameters for the meteorological variables.

| Variable | $\mu$ | $\sigma$ |
|---|---|---|
| BC (wet and dry) $(\mathrm{kgm^{-2}h^{-1}})$ | -2 | 1 |
| Dust (wet and dry) $(\mathrm{kgm^{-2}h^{-1}})$ | 0 | 1 |

**Table A2.** Perturbation parameters for the LAP fluxes.