# Peer review of "CrocO\_v1.0: a Particle Filter to assimilate snowpack observations in a spatialised framework"

_Geoscientific Model Development, 2020_

## Referee Comment (RC1) · Anonymous Referee #1 · 23 Aug 2020

In this study, the authors developed two variants of the particle filter (PF), named the global PF and the klocal PF, to assimilate snow depth and reflectance for snow water equivalent (SWE) estimation. The global PF assimilation all observations in the domain while the klocal PF is a localized PF that assimilate only a subset of observations. To prevent the degeneracy of PF, the global PF inflates the observation error covariance until a sufficient number of replicas are available, while the klocal approach applies the maximum of "k" observations to maintain a sufficiently large observation-state variable variation. Some notable assumptions include the observations are free of noise, error, and correlation in space and time, and the prior estimates and the observations are generated from the same model (identical twin). The results prove that the inflations and the k-localization effectively prevent the degeneracy, and the PF systems are able

to spread the observed snow signal to non-observed areas.

This is a nice contrition to the existing PF literature and has the potential to significantly extend the applicability of PF. The study fits the scope of the journal. I hope the authors consider the following comments in the revision:

1. The domain is divided into classes based on elevation band, aspect and slope, but there is no information regarding the geographic distribution of these classes. The PF's performance is generally good in high-elevation areas, but performance variations still exist among these areas. Could this be a result that the observation improve the more local classes more than the class that is farther away from the observation?

2. Some discussions on the assumption and the feasibility-testing nature of the system is needed in the abstract or be acknowledged in the introduction section. In addition to the assumptions mentioned above, the depth observation error is assumed to be 0.1m (error covariance is 1e-2m^2), which is quite a high-bar for existing observation techniques, especially when used on spaceborne platform for large-scale measurements.

3. Line 27: panel a of Figure 1 does not look like flat — the surface does seem to make an angle with the level surface (the brown triangle)

4. Line 128: it would be useful to include more details of the perturbation for each key forcing variable, like what perturbation models and error statistics are used, and whether spatial correlations are considered.

5. Figure 2: how do the forcing particle (Fi) and the model particle (Mi) get paired? Is it random or does it follow some protocol?

6. Line 180: can posterior estimates form the klocal approach show spatial discontinuity, since each area is updated independently by different measurements?

7. Line 195: how are the 10% and 0.3 here determined? Are they from previous literature or are there sensitivity test?

8 Line 239: the PF performance with band4 and band5 observations are quite different (as in Figure 4), what could be the reason?

9. Line 279: Figure 3c

10. Line 367: remove one "because of".

11. Figure 1: panel a is not "flat", as it has an elevation gradient. Making c the same size with b so their slope difference is more clear.
* * *
**[GMDD](https://doi.org)**

---

## Referee Comment (RC2) · Kristoffer Aalstad (Referee) · 26 Aug 2020

**General comments**

This manuscript presents a new ensemble-based snow data assimilation framework, Crocus-Observations (CrocO), to assimilate observations into the Crocus snowpack model in a semi-distributed geometry with a particle filter (PF). To address the issue of degeneracy, different variants of the PF are tested in a series of synthetic experiments where spatially sparse observations of height of snow (HS) or reflectance are assimilated for a massif (group of mountains) that is discretized into topographic classes. The sparsity of observations is meant to mimic the real situation where in-situ HS observations are usually only available for a handful of locations in a massif while clouds, shadows, and canopies can cause spatial gaps in (useful) reflectance retrievals. The objective is to use the PF to propagate information in space; i.e. to constrain the model ensemble not just in the observed classes, but also in the unobserved classes. The issue, compared to a completely local approach (called *rlocal*), is that this requires the assimilation of a larger number of observations which may trigger degeneracy. Through a series of 16 synthetic scenarios the authors demonstrate that it is possible to achieve such a propagation of information without degeneracy, both in the case of HS and reflectance assimilation, using either a global PF with inflation (called *global*) or a PF that is localized based on background correlations (called *klocal*).

This work fits well within the scope of GMD, and it is certainly of interest to the growing snow data assimilation community where the PF is gaining popularity. To my knowledge, it is also the first snow data assimilation study to demonstrate how the PF could be used in a spatialized context (non-local analyses) while avoiding degeneracy. The technical level of the work is also high with the framework being built up to eventually be run for operational purposes in an HPC environment. I therefore recommend this paper for publication pending minor revisions with a few technical concerns as outlined below.

**Specific comments**

L1 Consider changing *"the snowpack"* to just *"snowpack"* since not all snowpack properties are crucial.

L2 Change *"on the snowpack"* to *"on the state of the snowpack"*.

L4 Change *"inform on"* to *"provide information about"*.

L5 Change *"enables to estimate"* to *"enables the estimation of"*. It is not clear who or what is *"enabled to"*.

L7 Consider changing *"non observed"* to *"unobserved"*.

L10 Change *"known"* to *"prone"* and *"a too large number of"* to *"too many"*.

L34 It could be worth mentioning that higher resolution optical satellites (e.g. Landsat, Sentinel-2) are better able to resolve fractional snow cover at the MODIS scale (e.g. Aalstad et al., 2020, and references therein).

L38 Change *"enable to"* to (e.g.) *"enable us to"*.

L41 Change *"enables to"* to *"lets us"*.

L51-53 To be more precise I would suggest stating more explicitly that the two steps in the SIR PF analysis are importance sampling of the (unnormalized) posterior, with the prior as the proposal (or importance) density, followed by resampling to reduce the variance in the weights. In that way, it is also easier to understand the origin of the name "SIR". van Leeuwen (2009), who is already cited, explains these steps clearly for curious readers.

L55 When you say *"i.e. ..."* I expected a brief definition or explanation of what degeneracy is. Instead you state a consequence (or remedy) to degeneracy. It may be better to define degeneracy (as you do later on L163), after which you can mention

solutions. Moreover, degeneracy is only mentioned in the context of assimilating a large number of observations; which is seemingly what you try and deal with in this study. This problem can arise even in low dimensional states and is often a result of the likelihood (and thus posterior) becoming more peaked and harder to resolve with the available particles. An arguably broader issue that causes degeneracy with the PF (and importance sampling in general) is the curse of dimensionality where the required ensemble size (to avoid degeneracy) scales exponentially with the dimension of the state. This is also discussed in the studies of Synder et al. (2008); Bengtsson et al. (2008) that are already cited. I would suggest introducing the curse of dimensionality explicitly, since it can help explain why one expects that using a global (rather than local) PF algorithms, where the state space becomes much larger, is quite difficult. It is also surprising that the EnKF is barely mentioned, one of its strengths and the reason it is widely used in many applications is that it is more robust to this curse.

L60 While it is probably true that observation error variances are often underestimated, it is (in terms of Bayes' rule) strictly speaking incoherent to keep inflating these variances outside of certain frameworks such as likelihood tempering (see van Leeuwen et al., 2019, and references therein). Tempering of the likelihood explains the coherency of the ensemble smoother with multiple data assimilation (ES-MDA), used in Aalstad et al. (2018) for snow DA, which also inflates the observation error covariance matrix. It is not necessarily a big problem that the use of inflation here is incoherent, but the fact that it is a heuristic approach should be mentioned explicitly and potential solutions such as tempering could be proposed.

L69 Change *"It makes"* to *"This makes"*.

L79 Change *"operationally used"* to *"used operationally"*.

L81 Change *"enables to"* to *"enables us to"*.

L91 Change *"reflectance"* to *"reflectance observations"*.

L95 Change *"Following"* to *"Subsequently"*.

L101 Change *"the model into"* to *"the model for"*.

L102 Change *"enables to"* to *"enables us to"*.

L112 Change *"enabling to represent the snowpack coupling"* to *"coupling the snowpack with"*.

L117 Change *"This way,"* to *"As such,"*.

L140 In general I would suggest to put the hat just above the variable and not the sub/superscript. Similarly, I don't think sub/superscripts should be in bold since they are not matrices or vectors. That means (for example) using $\hat{x}^i_b$ rather than $\widehat{x^i_b}$ and $\hat{X}_b$ rather than $\widehat{X_b}$. This is a recurring issue throughout the math in the text. To conform with usual DA notation it might be better to not use a hat for the state (i.e. just $x$) and instead use a hat for the predicted observations ($\hat{x}$ or better yet $\hat{y}$).

L146 Remove *"supposed"* since you state the independence assumption in the ensuing brackets.

L148 Change *"type of variable of observation"* to *"type of observation"*.

L165 I didn't see $N_{eff}$ defined or even mentioned in Doucet et al. (2001), but maybe I missed it.

L168 Change *"sample population"* (a mix of distinct terms) to *"effective sample size"*.

L175 Change *"inspired on"* to *"inspired by"*.

L180 Change *"observations simultaneously assimilated"* to *"observations that are simultaneously assimilated"*.

L195&L201 I don't really follow the procedure here. First you say reflectance is not defined when there is no snow, then you say it is set to 0.2 for snow-free ground. Which is it? Are the bare ground reflectance values set as undefined or actually considered? I would expect the residuals to also contribute important information in the assimilation also in the cases that an observation or particle is bare as opposed to snow-covered.

L196 Why are *negative* background correlations considered "significant"? If the prior ensemble is negatively correlated between the analysis point and the observed point then surely the residuals (innovations) in the observed point should not necessarily be expected to carry over to the hypothetical residual at the analysis point? Is the reasoning that the hypothetical residual at the analysis point is in the perfectly negatively correlated case equal to minus the innovation at the observed point and that only the square of the innovation matters with a diagonal $\mathbf{R}$? Also, perhaps use

another term then *"significant"* which unfortunately still has strong statistical (null hypothesis significance testing) connotations.

L205 Change *"openloop"* to *"open-loop"*.

L206 The sentence *"These observations allow to mimic real observations with a perfect knowledge of the true state"* can easily be misunderstood to mean that real observations capture the true state. If anything, perfect observations are quite unrealistic and do not perfectly mimic reality at all. The fact that observations are not perfect is central to the Bayesian origins of ensemble-based DA in general and particle filtering in particular. With perfect observations DA just becomes an optimization problem. Ironically, you would end up with a sure-thing hypothesis (Jaynes, 2003; Schöniger et al., 2015), your likelihood would be a Dirac-delta function, and your particle weights would be nonsensical. In practice you do use a non-zero $\sigma_k^2$ in the analysis so this doesn't happen, but it is inconsistent to not perturb your synthetic observations.

L207 Change *"It allows"* to *"This allows us to"*.

L223 I guess by integral you really mean average? It is hard to imagine what the integral of SWE over time would represent physically unless it is normalized by the time period you are integrating over.

L224 On a first reading it was not clear why you extract percentiles of the open-loop ensemble to be used as synthetic observations. Perhaps you could make it clearer that you are effectively independently considering several different synthetic truth scenarios rather than a single truth run? Also, after you have extracted these different synthetic truth runs, what is in the way of perturbing the observed variables in these

(for each scenario) to generate synthetic observations as is usually done in twin experiments? This would allow for a more realistic evaluation, since real observations are noisy and you would still have access to the synthetic true SWE (unobserved) that you use in your evaluation?

L230 Change *"date"* to *"dates"*.

L233 Change *"is set"* to *"are set"*.

L235 Change *"uses only"* to *"only uses"*.

L247-265 When you compute your evaluation metrics you are using the corresponding truth not the corresponding (non-existent) observations. Your entire evaluation is based on how CrocO performs in terms of estimating the (unobserved) true SWE. As such, I suggest changing $o_{c,t}$ to $\mathcal{T}_{c,t}$ ($\mathcal{T}$ for truth, or something similar) and similarly for $O_{c,t}$ to make this clearer. Alternatively, you could be more explicit that all your evaluation is SWE-based and instead use notation like $SWE_{m,c,t}$ for the SWE ensemble and $SWE^{\star}_{c,t}$ for the true SWE in a given scenario?

L250 I suggest calling this the absolute error of the (ensemble) mean (AEM), to avoid confusion with the (ensemble) mean absolute error (MAE). For the caption of Figure 3, and when discussing this Figure (around L281) you call "AE" the RMSE which is incorrect. Judging by Fig.3a the RMSE would be considerably larger for the open-loop than for any of the analyses.

L264 This could be understood to mean that this is Eq. 8 in Hersbach (2000),

which it is not, and it is unusual to enumerate an equation (your Eq. 8) before it is presented on the next line. Furthermore, I couldn't find such an equation in Hersbach (2000), the closest I could find was his Eq. 39 which had an extra uncertainty term and a sign reversal for the "Resol" term. Could you explain the discrepancy?

L277 Change *"well representative"* to just *"representative"*.

L292 Change *"contrasted"* to *"contrasting"*.

L294 Change *"as for HS"* to just *"for HS"*?

L296 Again consider using another word than significant. Furthermore, are high background correlations that surprising given that, for a given ensemble member, you use the same multiphysics ($M_i$) and forcing perturbations ($F_i$) across the entire (semidistributed) domain? Isn't this mainly an indication that the SAFRAN forcing is quite spatially homogeneous (L128)?

L301 Change *"launched"* to *"conducted"*. In general, I would suggest referring to the SWE percentile-based sets of observations as *"synthetic observation scenarios"* rather than *"synthetic members"* to avoid confusion with the ensemble members.

L319 There are many examples in the literature of fractional snow-covered area (fSCA), which is retrieved from reflectance, constraining bulk variables like SWE quite well. HS observations are also often not representative of the model scale.

L320 Change *"all other things equal"* to *"all other things being equal"*. Perhaps make

it clearer that you are not jointly assimilating HS and reflectance in this experiment.

L324 Change *"well represent"* to *"properly represent"*. Also on the next line use (e.g.) *"marked"* instead of *"significant"*.

L327 Change *"with respect to"* to *"compared to"*.

L330 Why is *"Skill"* capitalized?

L350 I would recommend switching *"a right probability"* to *"the right frequency"*. Paraphrasing the discussion from the bottom of page 564 in Hersbach (2000): for the (average) CRPS, the reliability is similar to the rank histogram which can show if the frequency that the truth has a certain rank in the ensemble is equal for all ranks. In applications Bayesian (rather than frequentist) inference, which is what the PF is used for, there is an important distinction between the concept of frequency and probability; the latter is a measure of uncertainty (degree of belief, plausibility) (e.g. Lindley, 2000; Jaynes, 2003).

L356 Change *"conceptual"* to *"synthetic"*.

L360 Change *"on the"* to *"for the"*.

L362 This is an interesting speculation, but these are ensemble correlations between two areas in your domain not real spatial correlations. Maybe the ensemble is similar in the eastern and western aspects of the domain because a rain shadow effect (or something else) is not captured in your open-loop.

L364 Change *"such elevation"* to *"such elevations"*.

L370 I would argue that the fSCA depletion is quite informative for any seasonal snowpack, it is not necessarily maximally informative for intermittent snowpacks below the rain-snow line.

L372 Change *"well linked"* to *"closely linked"*.

L375 Change *"outstanding"* to *"unexpected"* and (next line) *"between these"* to "for these two".

L379 Change the sentence *"It is informative..."* to *"In our ensemble data assimilation framework, however, it does seem to be informative."*. On the next line I also recommend removing *"in this case"*.

L382 Change *"enabling to correct"* to *"enabling a correction of"*.

L387 Sentinel-2 and the Landsats should not be put in the same moderate resolution category as MODIS, VIIRS, and Sentinel-3.

L391 Change *"usually"* to *"often"* to qualify this statement.

L394-395 In terms of the current status of remote sensing of snow using optical satellites, this sentence seems too pessimistic. Even though Warren (2013) states that

retrieving BC content of snow from satellites is unlikely to be successful, it does not follow that reflectances retrievals from optical satellites are currently too inaccurate to be used to provide accurate information on snowpack properties. For example Aalstad et al. (2020) (and many other references therein) show that fractional snow-covered area (fSCA) can be estimated quite accurately from reflectances through a variety of methods using optical satellite sensors that are currently in orbit. These fSCA retrievals can, in turn, be used to constrain modeled estimates of other snowpack properties such as SWE through particle-based DA methods (see e.g. Alonso-González et al., 2020, for a recent example).

L408 Change *"informations"* to *"information"*.

L410 How can a correlation pattern based on an ensemble be realistic? In Bayesian inference the ensemble represents a probability distribution: a measure of uncertainty which is in the mind, not real. Jaynes (2003) explains this well with what he calls a mind projection fallacy: confusing reality and states of knowledge about reality.

L414 Change *"reliable model for that"* to *"reliable LAP model"*.

L418 Change *"suffers from obvious"* to *"suffering from obvious"* and *"suffer for large"* to *"suffer from large"*.

L421 As before, in Bayesian probability theory how can an ensemble correlation be real?

L424 Change *"area"* to *"areas"*.

L426 Change *"into larger"* to *"for larger"*.

L427 Change *"take the best"* to *"outperform"*.

L451 Change *"in the way of a new"* to just *"in a new"*.

L456 Change *"spatialized"* to *"semi-distributed"*. Also mention somewhere in the conclusion that this is a synthetic experiment.

L460 Capitalize the leading words in this enumeration.

L469 Change *"errors"* to *error"*.

L470 Again, why would fSCA only be worth assimilating at lower elevations? The depletion of fSCA might provide useful information anywhere in your domain. For example, Margulis et al. (2016) assimilated fSCA with a particle batch smoother (equivalent to your *rlocal* PF without resampling) to produce a 30 year high resolution snow reanalysis for the Californian Sierra Nevada with unprecedented accuracy. This study and others like it surely indicate that fSCA is quite valuable also for a PF even at higher elevations.

L490 Change *"softwares"* to *"software"*.

L500 Change *"enabling to"* to *"enabling us to"*.

[Figure]

Fig. 1 caption: Change *"elevation bands altitudes"* to *"altitudes of the elevation bands"*. Also change *"40° degrees slopes"* to *"40° slopes"* since the ° symbol is shorthand for degrees.

Fig. 2: Why is the superscript of the fourth prior particle at $t_1$ 3 and not 4? As suggested earlier for L140, consider changing the use of hats in your math notation.

Table 1: Change $N^*_{eeff}$ to $N^*_{eff}$. In the caption, change *"setup of"* to *"Setup for"* and change *"snow depth"* to *"height of snow"* to be consistent with the rest of the manuscript. The same applies to the title of subsection **4.2.1**.

Table 2: Change $N^*_{eeff}$ to $N^*_{eff}$. In the caption, change *"setup of"* to *"Setup for"*. Furthermore, change *"second"* to *"first"*; this is the first reflectance experiment.

Table 3: Same problems as with the other Tables.

Fig. 3: In the caption, change RMSE to AE (or AEM).

Fig. 4: In the caption, change *"the denote"* to *"denote the"*.

Fig. 4: In the caption, change *"on the whole"* to *"for the whole"*.

Fig. 6: In the caption, consider changing *"synthetic members"* to *"synthetic scenarios"* (since these are not ensemble members). Also, why is *"Skill"* capitalized?

Fig. 8&9: In the caption, consider changing *"member"* to *"scenario"* to avoid confusing the concept of your truth scenarios and the ensemble.

**References**

Aalstad et al.: Ensemble-based assimilation of fractional snow-covered area satellite retrievals to estimate the snow distribution at Arctic sites, TC, https://doi.org/10.5194/tc-12-247-2018, 2018.

Aalstad et al.: Evaluating satellite retrieved fractional snow-covered area at a high-Arctic site using terrestrial photography, RSE, https://doi.org/10.1016/j.rse.2019.111618, 2020.

Alonso-González et al.: Snowpack dynamics in the Lebanese mountains from quasi-dynamically downscaled ERA5 reanalysis updated by assimilating remotely-sensed fractional snow-covered area, HESSD, https://doi.org/10.5194/hess-2020-335, preprint under review, 2020.

Bengtsson et al.: Curse-of-dimensionality revisited: Collapse of the particle filter in very large scale systems, in: Probability and Statistics: Essays in Honor of David A. Freedman, https://doi.org/10.1214/193940307000000518, 2008.

Doucet et al.: An introduction to sequential Monte Carlo methods, in: Sequential Monte Carlo methods in practice, https://doi.org/10.1007/978-1-4757-3437-9_1, 2001.

Hersbach: Decomposition of the Continuous Ranked Probability Score for Ensemble Prediction Systems , WF, https://doi.org/10.1175/1520-0434(2000)015\%3C0559:DOTCRP\%3E2.0.CO;2, 2000.

Jaynes: Probability theory: The logic of science, https://doi.org/10.1017/CBO9780511790423, 2003.

Lindley: The philosophy of statistics, The Statistician, https://doi.org/10.1111/1467-9884.00238, 2000.

Margulis et al.: A Landsat-Era Sierra Nevada Snow Reanalysis (1985–2015), JHM, https://doi.org/10.1175/JHM-D-15-0177.1, 2016.

Schöniger et al.: A statistical concept to assess the uncertainty in Bayesian model weights and its impact on model ranking, WRR, https://doi.org/10.1002/2015WR016918, 2015.

Synder et al.: Obstacles to high-dimensional particle filtering, MWR, https://doi.org/10.1175/2008MWR2529.1, 2008.

van Leeuwen: Particle Filtering in Geophysical Systems, MWR, https://doi.org/10.1175/2009MWR2835.1, 2009.

van Leeuwen et al.: Particle filters for high-dimensional geoscience applications: A review, QJRMS, https://doi.org/10.1002/qj.3551, 2019.

Warren: Can black carbon in snow be detected by remote sensing?, JGR, https://doi.org/10.1029/2012JD018476, 2013.

---

## Author Comment (AC1) · 16 Nov 2020

GMD review

We would like to thank both referees for their extensive analysis of our manuscript which we believe helps a lot improving our paper. All the comments have been addressed and point by point response is provided below each comment. Note that some slight changes were made in the manuscript in order to improve its clarity, and are visible in the track changes. In the following, the reviewer initial comments are written in black, our answer in blue and the corrections in the paper are highlighted in red. Line references for modifications correspond to the initial submitted version of the manuscript, not the modified.

**Reviewer 1**

In this study, the authors developed two variants of the particle filter (PF), named the global PF and the klocal PF, to assimilate snow depth and reflectance for snow water equivalent (SWE) estimation. The global PF assimilation all observations in the domain while the klocal PF is a localized PF that assimilate only a subset of observations. To prevent the degeneracy of PF, the global PF inflates the observation error covariance until a sufficient number of replicas are available, while the klocal approach applies the maximum of "k" observations to maintain a sufficiently large observation-state variable variation. Some notable assumptions include the observations are free of noise, error, and correlation in space and time, and the prior estimates and the observations are generated from the same model (identical twin). The results prove that the inflations and the k-localization effectively prevent the degeneracy, and the PF systems are able to spread the observed snow signal to non-observed areas.

This is a nice contrition to the existing PF literature and has the potential to significantly extend the applicability of PF. The study fits the scope of the journal. I hope the authors consider the following comments in the revision:

The authors would like to thank Reviewer 1 for his/her thorough review and his/her questions on several subjects (the semi-distributed geometry, the methodology and assumptions, and the potential shortcomings of the different assimilation algorithms) which deserved more details and rigor in the formulation. We would like also to thank Reviewer 1 for expressing his/her need for more physical explanation on the ensemble correlation patterns of Fig. 4. We believe that these comments helped a lot improving the clarity of the manuscript, and we hope that the corrections fully address the reviewer comments.

1. The domain is divided into classes based on elevation band, aspect and slope, but there is no information regarding the geographic distribution of these classes. The PF's performance is generally good in high-elevation areas, but performance variations still exist among these areas. Could this be a result that the observation improve the more local classes more than the class that is farther away from the observation?

The semi-distributed framework does not allow to define a horizontal euclidian distance between topographic classes. Therefore, we do not consider any variability of the horizontal proximity between classes. However, in mountainous environments, topographic conditions often more directly drive snowpack variability than distance. As the reviewer points out, there is indeed a difference in performance between the observed classes and the unobserved classes, the former achieving better improvements, in general (see Figs. 6-7 and Sec. 4.2.1), and locations that are farther away (in model space) from the observations achieve the lowest performance (e.g. Fig. 8b 2100m, North, 40 degrees). Furthermore as the reviewer notes, there is also a notable variability of performance even among the observed classes, in particular for the reflectance assimilation.

According to this comment, the end of Sec. 4.2.2 (l. 331 of the manuscript) was amended to be more precise and descriptive:

Fig. 8. shows the spatial performance of the different algorithms for member 2016_p60. Spatial patterns similar to the HS assimilation are found. rlocal performance is limited to the observed classes, while global and klocal manage to improve the simulations across aspects and slopes. However, skill scores are lower than for HS (0.2-0.5), and the performance of all algorithms is poor in the classes that are the farther away from the observations, i.e. at lower elevations (600-900 m) and in some of the high altitude steep Northern classes (e.g. 2100_N_40 on Figs. 8b-c). Finally note that slight degradations of performance can sometimes be evidenced even in the observed classes for all the algorithms (e.g. in flat conditions at 3300 m on Fig. 8}a for the rlocal}, not evidenced by this example for the other algorithms).

2. Some discussions on the assumption and the feasibility-testing nature of the system is needed in the abstract or be acknowledged in the introduction section.
The feasibility-testing nature and the identical twin setup of this experiment were indeed not acknowledged enough in the abstract. This is now corrected on (L 13.14):
...based on background correlation patterns. Feasibility-testing experiments are carried out in an identical twin experiment setup, with synthetic observations of HS and VIS-NIR reflectances available in only a 1/6th of the simulation domain. …

Another notable assumption, the fact that observations are not corrupted, needed to be underlined and justified. We are actually conscious of this limitation, and a recent study has been submitted (Revuelto et al., submitted) in which we assimilate synthetic corrupted observations at the point scale. In our situation we did not corrupt the observations because little is known about the spatial structure of errors of reflectance (e.g. Cluzet et al., 2020): we know that assuming independent errors (i.e. diagonal R) is a very rough approximation of the reality which has strong consequences on the propagation of information. Corrupting the observations with such random structures would be theoretically more consistent, but would not yield much more insight on the potential for information from real observations to be spatially propagated as real spatial correlation of observation errors might be very different from this hypothesis. Future efforts should concentrate in better characterizing these spatial structures of errors. Consistently, the following sentence was modified at the beginning of Sec. 3 (L. 206)
Synthetic observations are extracted from a model run and assimilated without adding any noise. These observations mimic…

and a paragraph was added in the end of Sec. 5.2:
Regarding the observations, our study has some methodological limits, however. Observation errors are very roughly prescribed, and the assimilated observations are not corrupted as usually done in synthetic experiments (e.g. Durand et al., 2006). These choices were motivated by the fact that very little is known about the spatial correlation of reflectance observation errors in the semi-distributed setting (Cluzet et al., 2020). In a recently submitted paper, the impact of random and systematic errors of reflectance observations on point-scale assimilation experiments is thoroughly investigated (Revuelto et al., in prep). Efforts to better characterize these observation errors should be conducted in future work

Lastly, the synthetic nature of the experiments should be stated in the conclusions. The sentence on L490 was changed to:
In the framework of synthetic experiments, we have shown in particular that:…

In addition to the assumptions mentioned above, the depth observation error is assumed to be 0.1m (error covariance is 1e-2m^2), which is quite a high-bar for existing observation techniques, especially when used on space-borne platform for large-scale measurements.

We agree that the prescribed observation error is a high-bar for space-borne sensors. Indeed, results from recent studies such as Eberhard et al., (2020) could be used to provide a more accurate estimate of HS retrieval errors from satellites. Conversely, it could be considered as a low value for other sources of HS observations (e.g. stereo satellite imagery, Deschamps-Berger et al., 2020; local measurements with a high spatial representativeness error). As our work is a feasibility-testing experiment based on synthetic observations, an arbitrary observation error was chosen but indeed it may be important to adjust this value when applying the algorithm to real observations. This is now mentioned in the discussion on line 371:

Global and klocal algorithms exhibit strong performances when assimilating HS (Fig. 5). HS is closely linked with the SWE (by the bulk density) and the interest of this variable for data assimilation is clear (Margulis et al., 2019). Here, it should be kept in mind that HS assimilation is used as a baseline experiment to evaluate the algorithms and put reflectance assimilation into perspective. The prescribed HS observation errors ($\sigma_0=0.1m$) are not necessarily realistic. They should be adapted to the nature of the HS sensor. For example, space-borne HS observation errors are typically larger (e.g. Eberhard et al., 2020; Deschamps-Berger et al., 2020). The assimilation of such observations would probably yield lower improvements.

Though the performance is lower for Reflectance than in our HS experiments, it remains considerable and in line with previous results on point simulations (Charrois et al., 2016), with an average score improvement of 20-40\%…

Finally, note that the inflation procedure inside the global and rlocal approaches modifies the observation error which is assumed to be poorly known, reducing the impact of the prescribed value as mentioned in Sec. 2.3.1.

3. Line 27: panel a of Figure 1 does not look like flat — the surface does seem to make an angle with the level surface (the brown triangle)

This panel is actually flat, but we agreed the perspective view might be misleading. For this reason, we changed the background color of Fig. 1 in order to reinforce the perspective view, hoping that it helps.

Changed Fig. 1.

4. Line 128: it would be useful to include more details of the perturbation for each key forcing variable, like what perturbation models and error statistics are used, and whether spatial correlations are considered.

We agree tat this part was too elusive. Spatial correlations are not considered (i.e. equal to one) this is what we meant with "spatially homogeneous" (l. 128 of the manuscript), but this formulation could be misleading and more details were added.. For the sake of clarity, we also add the mention that perturbations are temporally correlated. The sentence was therefore modified accordingly

Before the beginning of the simulation, spatially homogeneous stochastic perturbations (e.g. at a given date, the same perturbation parameter is applied across the whole domain) with temporal auto-correlations are applied to this forcing to generate an ensemble of forcings.

In addition, an appendix was added giving more details on the perturbation procedure and parameters.

Added appendix A.

5. Figure 2: how do the forcing particle (Fi) and the model particle (Mi) get paired? Is it random or does it follow some protocol?

Yes, the pairing is random and keeps the same during the whole simulation. For the sake of clarity, the line 130-131 was changed to:

At the beginning of the simulation, each forcing $F_i$ is associated with a random $M_i$ ESCROC configuration and this relation is fixed during the whole simulation.

6. Line 180: can posterior estimates form the klocal approach show spatial discontinuity, since each area is updated independently by different measurements?

Thanks for underlining this point. Yes indeed klocalisation generates spatial discontinuities, it is one of the common drawbacks of localised approaches (see Farchi and Bocquet, 2018, already cited, for a thorough review). However, we expect the k-localisation to produce similar analyses (i.e. PF samples) for similar locations because their analyses will be based on similar sets of observations, thereby reducing the discontinuities compared to the r-local approach. In our setup, this has no direct consequence on the simulation as simulation points are independent, but it can hamper the interpretation of the spatial patterns of individual members. We changed the following lines inside the introduction (l. 69 of the manuscript):

It makes it possible to constrain the model in locations that are not directly observed, but with nearby observations. Contrary to global approaches, localisation has the disadvantage of producing spatially discontinuous analyses (each point receives a different analysis). This issue can be mitigated in various ways (Poterjoy, 2016; Farchi and Bocquet, 2018; Van Leeuwen et al., 2019). The underlying hypothesis…

Furthermore, we discussed this point in the end of Sec. 5.3:

Finally, in the case of modeled coupling between simulation points (e.g. snow drift), which was not the case here, the spatial discontinuities of the klocal analyses (see Sec. 1) might be a drawback compared to the global approach. Spatial discontinuities may reveal impractical for the interpretation of individual simulations outputs by snow forecasters too. The klocal approach is likely to reduce these discontinuities compared to the rlocal}, because similar locations will receive similar analyses (i.e. based on similar sets of observations). This issue could be partly mitigated by e.g. state-block-domain approaches (Farchi and Bocquet., 2018).

7. Line 195: how are the 10% and 0.3 here determined? Are they from previous literature or are there sensitivity test?

These parameters were adjusted during preliminary design experiments. As reflectance is not defined in the absence of snow, the number of pairs available to compute correlations between two locations varies for reflectance, and spurious high correlations are found when there is a very low number of common members. Regarding the 0.3 value, it is also an adjustment, based on the idea that if correlations are too low, it does not make sense to try to propagate information, as there will likely be a negative impact or no impact. The correlations exhibited on Fig. 4 enables the reader to realize typical (open-loop) correlation values with 40 members. We agree that a most rigorous definition based on significance levels would probably be a better option, and we will investigate this in future works. The following sentence on L196. was modified:

…, and match the following criteria: which were adjusted in preliminary experiments:
\begin{itemize}
\item in $\bm{x}^i_v$, there are at least 10\% of members defined in both points. As reflectance is not defined when there is no snow, spurious high correlations can be obtained when the computation of correlations is based on a very low number of pairs.
\item $\lvert \mathbf{B}_{\bm{v}}(n,p) \rvert >0.3$. If the absolute correlation is too low, it is likely that there is a poor potential for the distant observation to constrain the ensemble locally. In such a situation, it is better to reject the observation from the local analysis. Negative ensemble correlations can be physically sound, e.g. after a rain-on-snow event between the HS of two points separated by the rain-snow line. In such a situation, an HS observation on either point can hold information on precipitation rates at both locations. At the observed location, the PF will select the members with the

most appropriate precipitation rates. This sample is likely to perform well at both locations, so it can be used to constrain the unobserved location.
\end{itemize}

8 Line 239: the PF performance with band4 and band5 observations are quite different (as in Figure 4), what could be the reason?
 Note that Fig. 4 does not present the skill of an assimilation experiment, it is an example of open-loop ensemble background correlation patterns for band 4 and band 5 on a specific date. Regarding the interpretation of these results, there was a lack of physical explanations to help interpret the correlations of Band4 and Band5.  These observations are sensitive to the snowpack surface properties, namely the specific surface area (SSA, m²/kg) and light absorbing impurities content (LAP, g/g_snow). This is now stated in the introduction (line 30-31 of the manuscript):
For instance, snowpack VIS-NIR reflectances from moderate resolution (250-500 m) satellites  such as MODIS or Sentinel-3 can help constraining the snowpack surface properties such as microphysical properties (characterized by the specific surface area, SSA ($m^2kg^{-1}$) and light absorbing particles content (LAP, ($gg_{snow}^{-1}$)) (Durand et al., 2006; Dozier et al., 2009).

The individual sensitivity of the spectral reflectances is now further detailed (l. 232-233).
Reflectance is sensitive to the surface SSA and LAP (see Sec. 1). A minimal set of two different bands is used, corresponding to MODIS sensor band 4 (555 nm, sensitive to SSA and LAP) and 5 (1240 nm, mainly sensitive to SSA) (e.g. Fig. 2. of Cluzet et al., 2020).

A slight adjustment of the interpretation of Fig. 4 was performed to point negative correlations for Band5:
...being substantially correlated with the considered class. Note that negative correlations are evidenced with some lower altitude South-oriented topographic classes (e.g. 1500_S_40 on Fig. 4b). Finally, these patterns...

Indeed, the reasons why the correlation patterns of the different variables are different were already exposed in Sec. 5.2 & 5.3 but in a way too elusive way. This comment shows that the physical interpretation is very important to understand the paper and its motivations, and its absence might have been somewhat frustrating. In short (see track changes and Fig. 4): Band 4 is sensitive to SSA and LAP. LAP forcings are spatially uniform, partly explaining the rather constant and high spatial correlation of Band4. The spatial homogeneity of meteorological forcings also explains the strong HS correlations. Band 5 is sensitive to changes in surface micro-structural properties. Differential metamorphism can sometimes occur (between southern and northern aspect) causing a de-correlation in  band 5, potentially explaining what is observed on Fig. 4b. Negative correlations can also happen for the same reason between e.g. two elevations separated by the rain-snow line.

See the track-change throughout 5.2&5.3

Finally, investigating the skill of the PF as a function of the selected spectral bands is beyond the scope of this paper but note that this important topic is investigated by Revuelto et al., (submitted to Journal of Hydrology). This reference was clearly missing (because this reference was only in preparation when this manuscript was submitted).
We now refer to Revuelto et al., (submitted) in the last paragraph of Sec. 5.2.

9. Line 279: Figure 3c
Corrected
10. Line 367: remove one "because of".

Corrected
11. Figure 1: panel a is not "flat", as it has an elevation gradient.
We addressed this comment in the response to the referee's comment 3.
Making c the same size with b so their slope difference is more clear.
 We understand that the different horizontal extent between (b) and (c) might be confusing but in this schematic representation, it is important that  (a), (b) and (c) reach the same elevation. (c) appears smaller than (b) because it is steeper, but indeed they reach the same altitude. If (c) had the same basal area as (b) as suggested by the reviewer, it would have a similar size, but it would be twice as high, and unfortunately we believe that this would be detrimental to the description of the geometry.

**References**

Aalstad et al.: Ensemble-based assimilation of fractional snow-covered area satellite retrievals to estimate the snow distribution at Arctic sites, TC, https://doi.org/10.5194/tc-12-247-2018, 2018.

Aalstad et al.: Evaluating satellite retrieved fractional snow-covered area at a high-Arctic site using terrestrial photography, RSE, https://doi.org/10.1016/j.rse.2019.111618, 2020.

Alonso-González et al.: Snowpack dynamics in the Lebanese mountains from quasi-dynamically downscaled ERA5 reanalysis updated by assimilating remotely-sensed fractional snow-covered area, HESSD, https://doi.org/10.5194/hess-2020-335, preprint under review, 2020.

Bengtsson et al.: Curse-of-dimensionality revisited: Collapse of the particle filter in very large scale systems, in: Probability and Statistics: Essays in Honor of David A. Freedman, https://doi.org/10.1214/193940307000000518, 2008.

Candille, G., Brankart, J.-M., and Brasseur, P.: Assessment of an ensemble system that assimilates Jason-1/Envisat altimeter data in a probabilistic model of the North Atlantic ocean circulation., Ocean Science, 11, 425–438, 2015.

Charrois, L., Cosme, E., Dumont, M., Lafaysse, M., Morin, S., Libois, Q., and Picard, G.:
On the assimilation of optical reflectances and snow depth observations into a detailed
snowpack model, The Cryosphere, 10, 1021–1038, 10.5194/tc-10-1021-2016], 2016.

Cluzet, B., Revuelto, J., Lafaysse, M., Tuzet, F., Cosme, E., Picard, G., Arnaud, L., and Dumont, M.: Towards the assimilation of satellite reflectance into semi-distributed ensemble snowpack simulations, Cold Regions Science and Technology, 170, 102 918, 2020

De Lannoy, G. J., Reichle, R. H., Arsenault, K. R., Houser, P. R., Kumar, S., Verhoest, N. E., and Pauwels, V. R.: Multiscale assimilation of Advanced Microwave Scanning Radiometer–EOS snow water equivalent and Moderate Resolution Imaging Spectroradiometer snow cover fraction observations in northern Colorado, Water Resources Research, 48, 2012.

Doucet et al.: An introduction to sequential Monte Carlo methods, in: Sequential Monte Carlo methods in practice, https://doi.org/10.1007/978-1-4757-3437-9_1, 2001.

Durand, M. and Margulis, S. A.: Feasibility test of multifrequency radiometric data assimilation to estimate snow water equivalent, Journal of Hydrometeorology, 7, 443–457, 2006.

Eberhard, L. A., Sirguey, P., Miller, A., Marty, M., Schindler, K., Stoffel, A., and Bühler, Y.: Intercomparison of photogrammetric platforms for spatially continuous snow depth mapping, The Cryosphere Discussions, pp. 1–40, 2020.

Hersbach: Decomposition of the Continuous Ranked Probability Score for Ensemble Prediction Systems , WF, https://doi.org/10.1175/1520-0434(2000)015\%3C0559:DOTCRP\%3E2.0.CO;2, 2000.

Jaynes: Probability theory: The logic of science, https://doi.org/10.1017/CBO9780511790423, 2003.

Larue, F., Royer, A., Sève, D. D., Roy, A., and Cosme, E.: Assimilation of passive microwave AMSR-2 satellite observations in a snowpack evolution model over northeastern Canada, Hydrology and Earth System Sciences, 22, 5711–5734, 2018.

Leutbecher, M. and Haiden, T., Understanding changes of the continuous ranked probability score using a homogeneous gaussian approximation, QJRMS, 2020, 1-18

Lindley: The philosophy of statistics, The Statistician, https://doi.org/10.1111/1467-9884.00238, 2000.

Liu, J. S. and Chen, R.: Blind deconvolution via sequential imputations, Journal of the american statistical association, 90, 567–576, 1995.

Margulis et al.: A Landsat-Era Sierra Nevada Snow Reanalysis (1985–2015), JHM, https://doi.org/10.1175/JHM-D-15-0177.1, 2016.

Revuelto, J. et al. Assimilation of surface reflectance in snow simulations: impact on bulk snow variables (submitted to Journal of Hydrology).

Schöniger et al.: A statistical concept to assess the uncertainty in Bayesian model weights and its impact on model ranking, WRR, https://doi.org/10.1002/2015WR016918, 2015.

Snyder et al.: Obstacles to high-dimensional particle filtering, MWR, https://doi.org/10.1175/2008MWR2529.1, 2008.

van Leeuwen: Particle Filtering in Geophysical Systems, MWR, https://doi.org/10.1175/2009MWR2835.1, 2009.

van Leeuwen et al.: Particle filters for high-dimensional geoscience applications: A review, QJRMS, https://doi.org/10.1002/qj.3551, 2019.

Warren: Can black carbon in snow be detected by remote sensing?, JGR, https://doi.org/10.1029/2012JD018476, 2013.

---

## Author Comment (AC2) · 16 Nov 2020

We would like to thank both referees for their extensive analysis of our manuscript which we believe helps a lot improving our paper. All the comments have been addressed and point by point response is provided below each comment. Note that some slight changes were made in the manuscript in order to improve its clarity, and are visible in the track changes. In the following, the reviewer initial comments are written in black, our answer in blue and the corrections in the paper are highlighted in red. Line references for modifications correspond to the initial submitted version of the manuscript, not the modified.

**Reviewer 2 Kristoffer Aalstad**

**General comments**

This manuscript presents a new ensemble-based snow data assimilation framework, Crocus-Observations (CrocO), to assimilate observations into the Crocus snowpack model in a semi-distributed geometry with a particle filter (PF). To address the issue of degeneracy, different variants of the PF are tested in a series of synthetic experiments where spatially sparse observations of height of snow (HS) or reflectance are assimilated for a massif (group of mountains) that is discretized into topographic classes. The sparsity of observations is meant to mimic the real situation where in-situ HS observations are usually only available for a handful of locations in a massif while clouds, shadows, and canopies can cause spatial gaps in (useful) reflectance retrievals. The objective is to use the PF to propagate information in space; i.e. to constrain the model ensemble not just in the observed classes, but also in the unobserved classes. The issue, compared to a completely local approach (called rlocal), is that this requires the assimilation of a larger number of observations which may trigger degeneracy. Through a series of 16 synthetic scenarios the authors demonstrate that it is possible to achieve such a propagation of information without degeneracy, both in the case of HS and reflectance assimilation, using either a global PF with inflation (called global) or a PF that is localized based on background correlations (called klocal). This work fits well within the scope of GMD, and it is certainly of interest to the growing snow data assimilation community where the PF is gaining popularity. To my knowledge, it is also the first snow data assimilation study to demonstrate how the PF could be used in a spatialized context (non-local analyses) while avoiding degeneracy. The technical level of the work is also high with the framework being built up to eventually be run for operational purposes in an HPC environment. I therefore recommend this paper for publication pending minor revisions with a few technical concerns as outlined below.

The authors would like to thank Kristoffer Aalstad for this exhaustive review. We believe that comments helped to improve the clarity of several essential points (e.g. the formulation of the PF implementation, the statement of the degeneracy problem, and the non intuitive behavior of the PF in case of negative correlations). Regarding the motivations, and methodology, there was a lack of justification for the choice of the PF over the EnKF and obviously, the question of the SCF, an essential variable, was overlooked. Theoretical limitations of our work were shed to light, an issue which had to be acknowledged, even though we believe that we agree on the fact that it might not be severely detrimental to the applicability of our method. Finally, there were significant theoretical inputs on the bases of the PF and on potential avenues. These contributions were beneficial to the authors much beyond what will appear in the manuscript. The authors wanted to express their gratitude for this as well.

**Specific comments**

L1 Consider changing "the snowpack" to just "snowpack" since not all snowpack properties are crucial.

Corrected

L2 Change "on the snowpack" to "on the state of the snowpack".

Corrected

L4 Change "inform on" to "provide information about".

Corrected

L5 Change "enables to estimate" to "enables the estimation of". It is not clear who or what is "enabled to".

Corrected

L7 Consider changing "non observed" to "unobserved".

Thanks for the suggestion, changed throughout the text.

L10 Change "known" to "prone" and "a too large number of" to "too many".

Corrected

L34 It could be worth mentioning that higher resolution optical satellites (e.g. Landsat, Sentinel-2) are better able to resolve fractional snow cover at the MODIS scale (e.g. Aalstad et al., 2020, and references therein).

Thanks for this suggestion, this statement actually makes sense. Even though SCF is not the main focus in this work, we for sure consider assimilating it in future work, and it is worth mentioning it. In order not to loose track on our objectives (i.e. assimilating reflectances), we propose the following formulation, which acknowledges that SCF has more or less the same spatio-temporal limitations of reflectance, and mention that it saturates for deep snowpack.

The higher resolution offered by products from Landsat or Sentinel-2 might be an avenue to this issue (e.g. Masson et al., 2018; Aalstad et al., 2020) but at these resolution, reflectance retrievals are quite noisy due to e.g. digital elevation model errors (Cluzet et al., 2020). Finally, note that pixel fractional snow cover (snow cover fraction, SCF) can be accurately retrieved even from noisy reflectances (Sirguey et al., 2009; Aalstad et al., 2020), but it inherits the same spatio-temporal limitations as reflectances. SCF informativeness might also be limited in deep snowpack conditions (De Lannoy et al., 2012).

L38 Change "enable to" to (e.g.) "enable us to".

The sentence was changed to:

Detailed snowpack models are the only ones able to assess avalanche hazard and monitor water resources alike (Morin et al, 2020), but these applications are limited by their considerable errors and uncertainties (Essery et al., 2013; Lafaysse et al., 2017).

L41 Change "enables to" to "lets us".

The sentence was changed to:

Indeed, data assimilation combines the spatial and temporal coverage of snowpack models with the available information from observations in an optimal way.

L51-53 To be more precise I would suggest stating more explicitly that the two steps in the SIR PF analysis are importance sampling of the (unnormalized) posterior, with the prior as the proposal (or importance) density, followed by resampling to reduce the variance in the weights. In that way, it is also easier to understand the origin of the name "SIR". van Leeuwen (2009), who is already cited, explains these steps clearly for curious readers.

Thanks for this nice suggestion for improving this paragraph, explaining this two-step is a plus. However, as the referee understood, we are not familiar with Bayesian terms such as "importance sampling" and "proposal" and we are wondering whether using such terms would confuse readers without a background in Bayesian theory. We propose an alternative formulation, keeping the spirit of the two-steps and the term of "importance sampling", and helps understanding the "Sequential Importance Resampling" formulation.

The analysis of the PF-SIR (later on "PF") works in two steps. In a first step, so-called "importance sampling", the particles are weighted according to their distance to the observations (relative to the observation errors). Then, a resampling of the particles is performed in order to reduce the variance in the weights.

L55 When you say "i.e.. . . " I expected a brief definition or explanation of what degeneracy is. Instead you state a consequence (or remedy) to degeneracy. It may be better to define degeneracy (as you do later on L163), after which you can mention solutions.
We agree that the statement lacked rigor, it was reformulated. As this comment was separated in three and required nested modifications, please refer to the whole changes at the bottom of the whole comment.

Moreover, degeneracy is only mentioned in the context of assimilating a large number of observations; which is seemingly what you try and deal with in this study. This problem can arise even in low dimensional states and is often a result of the likelihood (and thus posterior) becoming more peaked and harder to resolve with the available particles. An arguably broader issue that causes degeneracy with the PF (and importance sampling in general) is the curse of dimensionality where the required ensemble size (to avoid degeneracy) scales exponentially with the dimension of the state. This is also discussed in the studies of Snyder et al. (2008); Bengtsson et al. (2008) that are already cited. I would suggest introducing the curse of dimensionality explicitly, since it can help explain why one expects that using a global (rather than local) PF algorithms, where the state space becomes much larger, is quite difficult.
This is a very interesting input, as it sheds light on the reasons why we expect the localised approach to be more suited to large scale problems, it was accounted for.

It is also surprising that the EnKF is barely mentioned, one of its strengths and the reason it is widely used in many applications is that it is more robust to this curse.
We agree that further discussion was needed on the PF vs. EnKF. Indeed, the main reason why we cannot consider using the EnKF is the Lagrangian formulation of our model which makes the computation of ensemble mean and updates impractical . This is thoroughly explained in Charrois et al. (2016) (already cited in the paper). On the contrary, as you say it is important to state that while applying the EnKF in spatialised application is quite easy, degeneracy/ curse of dimensionality are a severe drawbacks for the PF.

The concerned paragraph and the previous one were therefore modified:
The Particle Filter with sequential importance resampling (PF-SIR, Gordon 1993; van Leeuwen 2009 is a Bayesian ensemble data assimilation technique well suited to snowpack modeling (Dechant and Moradkhani, 2011; Charrois et al., 2016; Magnusson et al., 2017; Piazzi et al., 2018; Larue et al., 2018). The PF-SIR is a sequential algorithm relying on an ensemble of model runs (particles) which represents the forecast uncertainty. At each observation date, the prior (or background) composed of the particles is evaluated against the observations. The analysis of the PF-SIR (later on "PF") works in two steps. In a first step, so-called "importance sampling", the particles are weighted according to their distance to the observations (relative to the observation errors). Then, a resampling of the particles is performed in order to reduce the variance in the weights. The Ensemble Kalman Filter (EnKF, Evensen 2003), has also been widely used for snow cover data assimilation (e.g. Slater et al., 2006; De Lannoy et al., 2012; Magnusson et al., 2014). However, the PF is more adapted to models with a variable number of numerical layers such as detailed snowpack models (Charrois et al., 2016).
The PF could be used in a spatialised context to propagate the information from observation as suggested by Largeron et al., (2020) and Winstral et al., (2019). Contrary to the EnKF, such applications are rare to date (e.g. Thirel et al., 2013; Baba et al., 2018; Cantet et al., 2019). Indeed,

spatialised data assimilation with the PF is not straightforward because of the degeneracy issue, i.e. only a few particles are replicated in the analysis, often resulting in a poor representation of the forecast uncertainties. Degeneracy can be mitigated by increasing the number of particles, but the required population scales exponentially with the number of observations simultaneously assimilated (Snyder et al., 2008). Furthermore, an accurate representation of spatial error statistics by the ensemble is essential for the success of the assimilation system. To achieve that, the required ensemble size also scales exponentially with the system dimension, an issue known as the curse of dimensionality (Bengtsson, 2008). These issues are severe drawbacks when considering applications of the PF on large domains (i.e. implying a large number of observations and/or simulation points) with a reasonable number of particles (Stigter et al., 2017).

L60 While it is probably true that observation error variances are often underestimated, it is (in terms of Bayes' rule) strictly speaking incoherent to keep inflating these variances outside of certain frameworks such as likelihood tempering (see van Leeuwen et al., 2019, and references therein). Tempering of the likelihood explains the coherency of the ensemble smoother with multiple data assimilation (ES-MDA), used in Aalstad et al. (2018) for snow DA, which also inflates the observation error covariance matrix. It is not necessarily a big problem that the use of inflation here is incoherent, but the fact that it is a heuristic approach should be mentioned explicitly and potential solutions such as tempering could be proposed.

Thank you for the very interesting input on tempering methods. We didn't realize that Aalstad et al., (2018) was performing inflation, which we interpret as conceptually closer to the tempering presented in van Leeuwen et al., (2019) than to our approach. Our understanding of this literature and of the present comments is that tempering mitigates sampling issues but does not alter the extraction of information from observations since tempering/inflation coefficients sum to one. In contrary, our method does, and is therefore theoretically sub-optimal if not inconsistent. As you say, the inflation method we propose, as introduced by Larue et al., (2018) is a heuristic method aiming at mitigating mis-specified observation and representativeness errors. We acknowledge that this fact is worth underlining here (see changes). Meanwhile, we understand that tempering might be suited to tackle badly specified observation errors, but not in its present form. This is for sure an interesting lead to investigate. The following change is proposed for lines 58-62:

Several solutions exist to tackle the PF degeneracy. A first approach is to inflate the observation errors in the PF. The tolerance of the PF is increased, leading to more particles being replicated. This heuristic approach is based on the fact that observation error statistics (including sensor, retrieval and representativeness errors) are usually poorly known and underestimated. It can also be used as a safeguard to prevent the PF to degenerate on specific dates, when observations are not compatible with the ensemble. PF inflation was successfully implemented in point scale simulations of the snowpack (Larue et al., 2018).

see also change line 168 of the manuscript:

A first approach to mitigate degeneracy is to use inflation. This heuristic method iteratively...

L69 Change "It makes" to "This makes".
Corrected
L79 Change "operationally used" to "used operationally".
Corrected
L81 Change "enables to" to "enables us to".
Corrected
L91 Change "reflectance" to "reflectance observations".
Corrected
L95 Change "Following" to "Subsequently".
Corrected

L101 Change "the model into" to "the model for".

Corrected

L102 Change "enables to" to "enables us to".

Corrected

L112 Change "enabling to represent the snowpack coupling" to "coupling the snowpack with".

Corrected

L117 Change "This way," to "As such,".

Corrected

L140 In general I would suggest to put the hat just above the variable and not the sub/superscript. Similarly, I don't think sub/superscripts should be in bold since b ib rather than they are not matrices or vectors. That means (for example) using x*** x b and X b rather than X b . This is a recurring issue throughout the math in the text. To conform with usual DA notation it might be better to not use a hat for the state (i.e. just x) and instead use a hat for the predicted observations (x or better yet y)

Thanks for this rigorous input which has been accounted for in the revised manuscript.

L146 Remove "supposed" since you state the independence assumption in the
ensuing brackets.

Corrected

L148 Change "type of variable of observation" to "type of observation".

Corrected

L165 I didn't see N ef f defined or even mentioned in Doucet et al. (2001), but maybe I missed it.

Thanks for pointing this citation error. Correct reference is: Doucet, A.: On sequential simulation-based methods for Bayesian filtering, Tech. Rep., 1998, but it is not peer-reviewed, so we opted for Liu and Chen (1995)

L168 Change "sample population" (a mix of distinct terms) to "effective sample size".

Corrected

L175 Change "inspired on" to "inspired by".

Corrected

L180 Change "observations simultaneously assimilated" to "observations that are simultaneously assimilated".

Corrected

L195&L201 I don't really follow the procedure here. First you say reflectance is not defined when there is no snow, then you say it is set to 0.2 for snow-free ground. Which is it? Are the bare ground reflectance values set as undefined or actually considered? I would expect the residuals to also contribute important information in the assimilation also in the cases that an observation or particle is bare as opposed to snow-covered.

Thanks for this interesting remark which underlines the strong link between reflectance and SCF assimilation. We completely agree on the fact that snow/no snow holds precious information. Our choice was to not comment this question too much in order to focus on reflectances, but we agree it deserves clarifications. L.195 explains that TARTES optical scheme only provides snow reflectance (i.e. not a surface reflectance of a mixed soil-snow surface): this variable is not defined in the absence of snow. Some members being "undefined" is problematic for the PF. Conversely, in the observations, "no-snow" is an information, contrary to "no observation". For this reason, in  L201-202 we force a default value in the computation of the weights. By putting a reflectance of 0.2, (which corresponds to the bare soil broadband albedo in ISBA) in the unmasked snow-free synthetic observations and snow-free members, we extract this binary information in a very rough way. Ideally, future work should jointly assimilate reflectance and SCF in order to better leverage this information.
According to this explanation, Sec. 2.3.3 was expanded (the fact that reflectance observations are bounded was dropped):
Assimilating reflectance with the PF requires some adaptations. In Crocus, TARTES optical scheme (see Sec. 2.2.1) only provides snow reflectance, not all-surface reflectance: no value for the surface

reflectance is issued in the absence of snow. Conversely, the weights of the particles are not defined in Eq. 2 if the members are snow-free. These issues were roughly accommodated by setting the reflectances of snow-free members and observations to 0.2 (the value of bare soil broadband albedo in ISBA model) in the PF Eq. 2 (Sec. 2.2.3).

L196 Why are negative background correlations considered "significant"? If the prior ensemble is negatively correlated between the analysis point and the observed point then surely the residuals (innovations) in the observed point should not necessarily be expected to carry over to the hypothetical residual at the analysis point? Is the reasoning that the hypothetical residual at the analysis point is in the perfectly negatively correlated case equal to minus the innovation at the observed point and that only the square of the innovation matters with a diagonal R?

Thanks for pointing the question of negative correlations. We are not sure to fully understand the question, so we try to answer but we might have missed something. Negative correlations can be physically sound. Consider HS and two points separated by the rain-snow line during a rain-on-snow event, an ensemble built by perturbations on the precipitation rates, and an observation available after the precipitation event. In the snowy (rainy) point, the members with the highest solid (liquid) precipitation will see their HS increase (decrease), resulting in a negative ensemble correlation between the HS of the two points. Now consider that only the HS of the snowy point is observed, and that the ensemble underestimated HS: it is likely that precipitation rates were underestimated at both locations: HS is likely overestimated in the rainy point. The PF will select the members with the highest precipitation rates at the snowy point, but this information is also valid for the rainy point, and therefore this information should be transferred by using the same PF sample there.

The correspond item was therefore modified:

\item $\lvert \mathbf{B}_{\bm{v}}(n,p) \rvert >0.3$. If the absolute correlation is low, it is likely that there is a poor potential for the distant observation to constrain the ensemble locally. In such a situation, it is better to reject the observation from the local analysis. Negative ensemble correlations can be physically sound, e.g. after a rain-on-snow event between the HS of two points separated by the rain-snow line. In such a situation, an HS observation on either point can hold information on precipitation rates at both locations. At the observed location, the PF will select the members with the most appropriate precipitation rates. This sample is likely to perform well at both locations, so it can be used to constrain the unobserved location.

Also, perhaps use another term then "significant" which unfortunately still has strong statistical (null hypothesis significance testing) connotations.

Thanks for this remark, this was modified throughout the text.

L205 Change "openloop" to "open-loop".

Corrected (multiple changes).

L206 The sentence "These observations allow to mimic real observations with a perfect knowledge of the true state" can easily be misunderstood to mean that real observations capture the true state. If anything, perfect observations are quite unrealistic and do not perfectly mimic reality at all. The fact that observations are not perfect is central to the Bayesian origins of ensemble-based DA in general and particle filtering in particular. With perfect observations DA just becomes an optimization problem. Ironically, you would end up with a sure-thing hypothesis (Jaynes, 2003; Schöniger et al., 2015), your likelihood would be a Dirac-delta function, and your particle weights would be nonsensical. In practice you do use a non-zero σ k 2 in the analysis so this doesn't happen, but it is inconsistent to not perturb your synthetic observations.

Thanks for his thorough remark. Despite this is  mentioned on L209, we acknowledge that the fact that we don't corrupt the observations should be pointed out more clearly as a limit of our methodological

study compared to the literature (e.g. Durand et al., 2006) despite recent studies did not do so either (e.g. Charrois et al., 2016). We are actually conscious of this limitation, and a recent study has been submitted (Revuelto et al., submitted) in which we assimilate synthetic corrupted observations at the point scale. In our situation we did not corrupt the observations because little is known about the spatial structure of errors of reflectance (e.g. Cluzet et al., 2020): we know that assuming independent errors (i.e. diagonal R) is a very rough approximation of the reality which has strong consequences on the propagation of information. Corrupting the observations with such random structures would be theoretically more consistent, but would not yield much more insight on the potential for information from real observations to be spatially propagated as real spatial correlation of observation errors might be very different from this hypothesis. Future efforts should concentrate in better characterizing these spatial structures of errors. Consistently, the following sentence was modified:

Synthetic observations are extracted from a model run and assimilated without adding any noise. These observations mimic...

and a paragraph was added in the end of Sec. 5.2:

Regarding the observations, our study has some methodological limits, however. Observation errors are very roughly prescribed, and the assimilated observations are not corrupted as usually done in synthetic experiments (e.g. Durand et al., 2006). These choices were motivated by the fact that very little is known about the spatial correlation of reflectance observation errors in the semi-distributed setting (e.g. Cluzet et al., 2020). In a recently submitted paper, the impact of random and systematic errors of reflectance observations on point-scale assimilation experiments is thoroughly investigated (Revuelto et al., 2021). Efforts to better characterize these observation errors should be conducted in future work

L207 Change "It allows" to "This allows us to". Linked to previous comment.
Corrected

L223 I guess by integral you really mean average? It is hard to imagine what the integral of SWE over time would represent physically unless it is normalized by the time period you are integrating over.
Indeed we computed the time integral, for the sake of computational simplicity, not the average. There is only a proportionality factor between the integral and the average, so SWE percentiles correspond to average SWE percentiles. We propose to simplify the statement by replacing the "integral" by "average", which makes it more sound, and does not change anything to the idea.
Changed "integral" to "average"

L224 On a first reading it was not clear why you extract percentiles of the open-loop ensemble to be used as synthetic observations. Perhaps you could make it clearer that you are effectively independently considering several different synthetic truth scenarios rather than a single truth run?
Thanks for pointing out this lack of clarity. The following sentence was added on L225:
...e.g. 2014_p80). This method enables us to evaluate the efficiency of data assimilation experiments under contrasted snow condition scenarios. Before any assimilation experiment...

Also, after you have extracted these different synthetic truth runs, what is in the way of perturbing the observed variables in these (for each scenario) to generate synthetic observations as is usually done in twin experiments? This would allow for a more realistic evaluation, since real observations are noisy and you would still have access to the synthetic true SWE (unobserved) that you use in your evaluation?
See previous answer to comment from L206.
L230 Change "date" to "dates".
Corrected
L233 Change "is set" to "are set".
Corrected

L235 Change "uses only" to "only uses".

Corrected

L247-265 When you compute your evaluation metrics you are using the corresponding truth not the corresponding (non-existent) observations. Your entire evaluation is based on how CrocO performs in terms of estimating the (unobserved) true SWE. As such, I suggest changing o c,t to T c,t (T for truth, or something similar) and similarly for O c,t to make this clearer. Alternatively, you could be more explicit that all your evaluation is SWE-based and instead use notation like SW E m,c,t for the SWE ensemble and SW E ?c,t for the true SWE in a given scenario?

Thanks for this nice suggestion. We opted for the first option, substituting o_{c,t by \tau_{c,t}. (see changes).

L250 I suggest calling this the absolute error of the (ensemble) mean (AEM), to avoid confusion with the (ensemble) mean absolute error (MAE).

Thanks for this comment. We opted for the AEM name, which is unambiguous. Modifications were performed accordingly (including Fig. 3.)

For the caption of Figure 3, and when discussing this Figure (around L281) you call "AE" the RMSE which is incorrect. Judging by Fig.3a the RMSE would be considerably larger for the open-loop than for any of the analyses.

Thanks for this comment. We actually forgot to replace RMSE by AE (AEM) in the text, thanks for pointing this out !

L264 This could be understood to mean that this is Eq .8 in Hersbach (2000),which it is not, and it is unusual to enumerate an equation (your Eq. 8) before it is presented on the next line. Furthermore, I couldn't find such an equation in Hersbach (2000), the closest I could find was his Eq. 39 which had an extra uncertainty term and a sign reversal for the "Resol" term. Could you explain the discrepancy?

Thanks for pointing this out. There was an error in the reference, the appropriate one being Candille et al., 2015. While the interpretation of the Reli term is unambiguous, interpretation of the Resol term is more controversial (P. de Mey and G. Candille, personal communication). This is why we didn't focus on the Resol term. Recent publication from Leutbecher et al., (2020) might help understanding Resol for curious readers.

L277 Change "well representative" to just "representative".

Corrected

L292 Change "contrasted" to "contrasting".

Corrected

L294 Change "as for HS" to just "for HS"?

, we actually mean that for band4, spatial correlation patterns are similar to those of HS.

We replaced "as for HS" by "Similar results are obtained for HS."

L296 Again consider using another word than significant.

Corrected

Furthermore, are high background correlations that surprising given that, for a given ensemble member, you use the same multiphysics (M i ) and forcing perturbations (F i ) across the entire (semidistributed) domain? Isn't this mainly an indication that the SAFRAN forcing is quite spatially homogeneous (L128)?

Thanks for this remark. Indeed, intrinsic correlations come from the forcing variables and ESCROC members and, this point is actually discussed in Sec 5.3 of the manuscript. Nevertheless Fig.4a-b actually shows that despite same Mi and Fi are applied across the entire domain, some locations are completely decorrelated due to the combination of strong vertical gradients and some highly non-linear processes.

We amended L412 of the manuscript to also mention that ESCROC members were also spatially constant:

Strong Band 4 correlations might be due to the spatially homogeneous perturbations of LAP fluxes used to force the simulations (see Sec. 2.2.2), a key driver of this variable, and because the same snow model configuration is applied for a given member across the simulation domain. Several studies suggest...

Moreover, this comment points out a lack of interpretation of these background correlations in our manuscript. In line with our answer to comment 8 of Reviewer 1, physical interpretation of the Band 5 background correlations evidenced in Fig. 4b is a bit more complex than for Band 4 and HS. Differential metamorphism can happen between the opposite sides of a mountain (because of different illumination conditions), or similarly, across the rain snow line, resulting in a de-correlation of band 5 reflectances. Details on these processes were added throughout sections 5.2 and 5.3, and we suggest to refer to the track-change.
See track change in Secs. 5.2 and 5.3

L301 Change "launched" to "conducted". In general, I would suggest referring to the SWE percentile-based sets of observations as "synthetic observation scenarios" rather than "synthetic members" to avoid confusion with the ensemble members.
Corrected, thanks for this suggestion.
L319 There are many examples in the literature of fractional snow-covered area (fSCA), which is retrieved from reflectance, constraining bulk variables like SWE quite well.
We agree and following your suggestion, several significant changes been made throughout the text (introduction discussion and conclusions, see in particular the answer to L.34, L. 391, L470. We hope these corrections are enough. Here, for the sake of clarity, we propose to correct to:
"raw reflectance products"

HS observations are also often not representative of the model scale.
This is an important point on which we completely agree, but we do not aim at discussing HS assimilation too much as the main focus of this study is reflectance.

L320 Change "all other things equal" to "all other things being equal". Perhaps make it clearer that you are not jointly assimilating HS and reflectance in this experiment.
Thanks for this suggestion. The sentence was changed to:
In order to assess this difference, we conduct assimilation of reflectance only, in the same setup as in Sec. 4.2.1}, all other things being equal.

L324 Change "well represent" to "properly represent". Also on the next line use (e.g.) "marked" instead of "significant".
Corrected
L327 Change "with respect to" to "compared to".
Corrected
L330 Why is "Skill" capitalized?
Corrected
L350 I would recommend switching "a right probability" to "the right frequency". Paraphrasing the discussion from the bottom of page 564 in Hersbach (2000): for the (average) CRPS, the reliability is similar to the rank histogram which can show if the frequency that the truth has a certain rank in the ensemble is equal for all ranks. In applications Bayesian (rather than frequentist) inference, which is what the PF is used for, there is an important distinction between the concept of frequency and probability; the latter is a measure of uncertainty (degree of belief, plausibility) (e.g. Lindley, 2000; Jaynes, 2003).

Corrected, accounted for, thanks a lot for this input.

L356 Change "conceptual" to "synthetic".
Corrected
L360 Change "on the" to "for the".
Corrected
L362 This is an interesting speculation, but these are ensemble correlations between two areas in your domain not real spatial correlations. Maybe the ensemble is similar in the eastern and western aspects of the domain because a rain shadow effect (or something else) is not captured in your open-loop.
The potential mismatch between ensemble correlations and real spatial correlations is discussed in L411 of the manuscript. We agree that as you mention, not accounting for the intra-massif variability of meteorological conditions (in the sense that you mean, e.g. Western slopes should lie preferentially in the windward side of the massif and receive more/less snow than those on the windward side). On L. 411, we added a mention to this:
Strong and almost uniform HS correlations (Fig. 4b) might be caused by the satial homogeneity of precipitation perturnations and because we do not account for e.g. wind drift, intra-massif variability of meteorological conditions and gravitational redistribution of snow (Wayand et al., 2018)....
L364 Change "such elevation" to "such elevations".
Corrected
L370 I would argue that the fSCA depletion is quite informative for any seasonal snowpack, it is not necessarily maximally informative for intermittent snowpacks below the rain-snow line.
Thanks again for this remark. We believe that answer to L319 comment is appropriate here too.
L372 Change "well linked" to "closely linked".
Corrected
L375 Change "outstanding" to "unexpected" and (next line) "between these" to "for these two".
Corrected, the sentence was changed to:
This study unexpectedly suggests that reflectance information can be spread from southern slopes to the northern ones,...

L379 Change the sentence "It is informative. . . " to "In our ensemble data assimilation framework, however, it does seem to be informative.". On the next line I also recommend removing "in this case".
Corrected
L382 Change "enabling to correct" to "enabling a correction of".
Corrected, changed to: … parametrisations, thus correcting the ensemble...
L387 Sentinel-2 and the Landsats should not be put in the same moderate resolution category as MODIS, VIIRS, and Sentinel-3.
Corrected, changed to: …the abundance of moderate to high resolution space-borne sensors (MODIS, Sentinel2-3, VIIRS, Landsat…)…

L391 Change "usually" to "often" to qualify this statement.
Corrected, changed to : "generally" a bit stronger than "often".

L394-395 In terms of the current status of remote sensing of snow using optical satellites, this sentence seems too pessimistic. Even though Warren (2013) states that retrieving BC content of snow from satellites is unlikely to be successful, it does not follow that reflectances retrievals from optical satellites are currently too inaccurate to be used to provide accurate information on snowpack properties.For example Aalstad et al. (2020) (and many other references therein) show that fractional snow-covered area (fSCA) can be estimated quite accurately from reflectances through a variety of methods using optical satellite sensors that are currently in orbit. These fSCA retrievals can, in turn, be

used to constrain modeled estimates of other snowpack properties such as SWE through particle-based DA methods (see e.g. Alonso-González et al., 2020, for a recent example).

Thanks for pointing out this sentence whose formulation was inappropriate. Our purpose was to talk about surface properties (grain size, and light absorbing particle contents (LAP)). Thanks to a previous comment (L34), it is now acknowledged that SCF is accurately retrieved. Regarding surface properties, as you say, Warren (2013) statement only stands for LAP, while for snow microphysical properties, the required accuracy might be reached. And of course, for SCF, it is already the case, this point was added in the introduction (see changes to comment on line 34).

In the near-infrared, the signal-to-noise ratio of reflectances observations might be sufficient to constrain the surface microphysical properties (Durand et al., 2007; Mary et al., 2013), whereas the required accuracy for visible reflectance retrievals to remain informative on the snowpack light absorbing particles content is high (Warren, 2013), and it is yet to prove whether either approach can achieve this requirement.

Regarding the interest of SCF for data assimilation, we agree also on its added value, and that it needed to be acknowledged in the discussion. A sentence was added at the beginning paragraph of Sec. 5.3 (L. 388)

Reflectance is an appealing variable for snowpack modelling because of its sensitivity to snowpack surface properties (Dozier, 2009) and the abundance of moderate to high resolution space-borne sensors (MODIS, Sentinel2-3, VIIRS, Landsat...) providing us with a handful of observations to assimilate, contrary to HS. The potential for assimilation of SCF, which is retrieved from reflectances, is clear (Margulis et al., 2016, Aalstad et al., 2018, Alonso-Gonzalez et al., 2020). This study demonstrates the potential of the PF to spread information and assimilate raw reflectances with a positive impact (Sec. 5.2). Yet, assimilating real observations of reflectance is another challenge, for two reasons.

L408 Change "informations" to "information".
Corrected

L410 How can a correlation pattern based on an ensemble be realistic? In Bayesian inference the ensemble represents a probability distribution: a measure of uncertainty which is in the mind, not real. Jaynes (2003) explains this well with what he calls a mind projection fallacy: confusing reality and states of knowledge about reality.

We acknowledge this is a bad formulation, thank you for pointing this out. We mean: based on the assumption that ensemble background correlations are a realistic representation of modeling errors. The sentence was changed to:

The klocal algorithm could be more suited to this situation, because it is looking for local optima, based on the assumption that background correlation are a realistic representation of modelling errors.

L414 Change "reliable model for that" to "reliable LAP model".
Corrected, changed to : ...no reliable model of such processes exists in complex terrain.

L418 Change "suffers from obvious" to "suffering from obvious" and "suffer for large" to "suffer from large".
Corrected

L421 As before, in Bayesian probability theory how can an ensemble correlation be real?
See answer to L410: changed to:

In the future, improving the ability of ensemble correlations to represent modelling errors correlations could make the spreading of information an even more challenging task with the klocal algorithm.

L424 Change "area" to "areas".

Corrected

L426 Change "into larger" to "for larger".

Corrected

L427 Change "take the best" to "outperform".

Corrected

L451 Change "in the way of a new" to just "in a new".

Corrected

L456 Change "spatialized" to "semi-distributed".

We acknowledge that this work has only been done in a semi-distributed geometry, which is a spatialized setting. We would like to stick with the use of "spatialized" because the term "semi-distributed" is quite obscure for the majority of the people, and it might confuse the audience especially if they only read the abstract/conclusion. We consider that it is clearly stated everywhere else in the paper that we work in a semi-distributed setting and that this will be clear for curious readers. Finally, as we mentioned later on, nothing specific to the semi-distrubited geometry was developed here: CrocO can be applied seamlessly on networks of in-situ stations and fully distributed frameworks.

Also mention somewhere in the conclusion that this is a synthetic experiment.

We completely agree that this should be mentioned. The sentence on L490 was changed to:

In the framework of synthetic experiments, we have shown in particular that:

L460 Capitalize the leading words in this enumeration.

Corrected

L469 Change "errors" to error".

Corrected

L470 Again, why would fSCA only be worth assimilating at lower elevations? The depletion of fSCA might provide useful information anywhere in your domain. For example, Margulis et al. (2016) assimilated fSCA with a particle batch smoother (equivalent to your rlocal PF without resampling) to produce a 30 year high resolution snow reanalysis for the Californian Sierra Nevada with unprecedented accuracy. This study and others like it surely indicate that fSCA is quite valuable also for a PF even at higher elevations.

We agree that this statement was too pessimistic regarding the SCF. The sentence was changed to:

Snow cover fraction would be a good companion variable to jointly assimilate with reflectances, requiring the use of an appropriate observation operator.

L490 Change "softwares" to "software".

Corrected

L500 Change "enabling to" to "enabling us to".

Corrected, changed to: necessary to

Fig. 1 caption: Change "elevation bands altitudes" to "altitudes of the elevation bands". Also change "40 ◦ degrees slopes" to "40 ◦ slopes" since the ◦ symbol is shorthand for degrees.

Corrected

Fig. 2: Why is the superscript of the fourth prior particle at t 1 3 and not 4? As suggested earlier for L140, consider changing the use of hats in your math notation.

Corrected

Table 1: Change N eeff to N ef f . In the caption, change "setup of" to "Setup for" and change "snow depth" to "height of snow" to be consistent with the rest of themanuscript. The same applies to the title of subsection 4.2.1.

Corrected

Table 2: Change N eeff to N eff . In the caption, change "setup of" to "Setup for". Furthermore, change "second" to "first"; this is the first reflectance experiment.

Corrected

Table 3: Same problems as with the other Tables.
Corrected
Fig. 3: In the caption, change RMSE to AE (or AEM).
Corrected, changed to AEM
Fig. 4: In the caption, change "the denote" to "denote the".
Corrected
Fig. 4: In the caption, change "on the whole" to "for the whole".
Corrected
Fig. 6: In the caption, consider changing "synthetic members" to "synthetic scenarios" (since these are not ensemble members). Also, why is "Skill" capitalized?
Corrected accounted for, according to previous corrections.
Fig. 8&9: In the caption, consider changing "member" to "scenario" to avoid confusing the concept of your truth scenarios and the ensemble.
Corrected

**References**

Aalstad et al.: Ensemble-based assimilation of fractional snow-covered area satellite retrievals to estimate the snow distribution at Arctic sites, TC, https://doi.org/10.5194/tc-12-247-2018, 2018.

Aalstad et al.: Evaluating satellite retrieved fractional snow-covered area at a high-Arctic site using terrestrial photography, RSE, https://doi.org/10.1016/j.rse.2019.111618, 2020.

Alonso-González et al.: Snowpack dynamics in the Lebanese mountains from quasi-dynamically downscaled ERA5 reanalysis updated by assimilating remotely-sensed fractional snow-covered area, HESSD, https://doi.org/10.5194/hess-2020-335, preprint under review, 2020.

Bengtsson et al.: Curse-of-dimensionality revisited: Collapse of the particle filter in very large scale systems, in: Probability and Statistics: Essays in Honor of David A. Freedman, https://doi.org/10.1214/193940307000000518, 2008.

Candille, G., Brankart, J.-M., and Brasseur, P.: Assessment of an ensemble system that assimilates Jason-1/Envisat altimeter data in a probabilistic model of the North Atlantic ocean circulation., Ocean Science, 11, 425–438, 2015.

Charrois, L., Cosme, E., Dumont, M., Lafaysse, M., Morin, S., Libois, Q., and Picard, G.:
On the assimilation of optical reflectances and snow depth observations into a detailed
snowpack model, The Cryosphere, 10, 1021–1038, 10.5194/tc-10-1021-2016], 2016.

Cluzet, B., Revuelto, J., Lafaysse, M., Tuzet, F., Cosme, E., Picard, G., Arnaud, L., and Dumont, M.: Towards the assimilation of satellite reflectance into semi-distributed ensemble snowpack simulations, Cold Regions Science and Technology, 170, 102 918, 2020

De Lannoy, G. J., Reichle, R. H., Arsenault, K. R., Houser, P. R., Kumar, S., Verhoest, N. E., and Pauwels, V. R.: Multiscale assimilation of Advanced Microwave Scanning Radiometer–EOS snow water equivalent and Moderate Resolution Imaging Spectroradiometer snow cover fraction observations in northern Colorado, Water Resources Research, 48, 2012.

Doucet et al.: An introduction to sequential Monte Carlo methods, in: Sequential Monte Carlo methods in practice, https://doi.org/10.1007/978-1-4757-3437-9_1, 2001.

Durand, M. and Margulis, S. A.: Feasibility test of multifrequency radiometric data assimilation to estimate snow water equivalent, Journal of Hydrometeorology, 7, 443–457, 2006.

Eberhard, L. A., Sirguey, P., Miller, A., Marty, M., Schindler, K., Stoffel, A., and Bühler, Y.: Intercomparison of photogrammetric platforms for spatially continuous snow depth mapping, The Cryosphere Discussions, pp. 1–40, 2020.

Hersbach: Decomposition of the Continuous Ranked Probability Score for Ensemble Prediction Systems , WF, https://doi.org/10.1175/1520-0434(2000)015\%3C0559:DOTCRP\%3E2.0.CO;2, 2000.

Jaynes: Probability theory: The logic of science, https://doi.org/10.1017/CBO9780511790423,

2003.

Larue, F., Royer, A., Sève, D. D., Roy, A., and Cosme, E.: Assimilation of passive microwave AMSR-2 satellite observations in a snowpack evolution model over northeastern Canada, Hydrology and Earth System Sciences, 22, 5711–5734, 2018.

Leutbecher, M. and Haiden, T., Understanding changes of the continuous ranked probability score using a homogeneous gaussian approximation, QJRMS, 2020, 1-18

Lindley: The philosophy of statistics, The Statistician, https://doi.org/10.1111/1467-9884.00238, 2000.

Liu, J. S. and Chen, R.: Blind deconvolution via sequential imputations, Journal of the american statistical association, 90, 567–576, 1995.

Margulis et al.: A Landsat-Era Sierra Nevada Snow Reanalysis (1985–2015), JHM, https://doi.org/10.1175/JHM-D-15-0177.1, 2016.

Revuelto, J. et al. Assimilation of surface reflectance in snow simulations: impact on bulk snow variables (submitted to Journal of Hydrology).

Schöniger et al.: A statistical concept to assess the uncertainty in Bayesian model weights and its impact on model ranking, WRR, https://doi.org/10.1002/2015WR016918, 2015.

Snyder et al.: Obstacles to high-dimensional particle filtering, MWR, https://doi.org/10.1175/2008MWR2529.1, 2008.

van Leeuwen: Particle Filtering in Geophysical Systems, MWR, https://doi.org/10.1175/2009MWR2835.1, 2009.

van Leeuwen et al.: Particle filters for high-dimensional geoscience applications: A review, QJRMS, https://doi.org/10.1002/qj.3551, 2019.

Warren: Can black carbon in snow be detected by remote sensing?, JGR, https://doi.org/10.1029/2012JD018476, 2013.